# kNN-Diffusion: Image Generation via Large-Scale Retrieval

**Shelly Sheynin**[*], **Oron Ashual**[*],
**Adam Polyak**, **Uriel Singer**, **Oran Gafni**, **Eliya Nachmani**, **Yaniv Taigman**
[*]Equal Contribution               Meta AI
{shellysheynin,oron}@meta.com

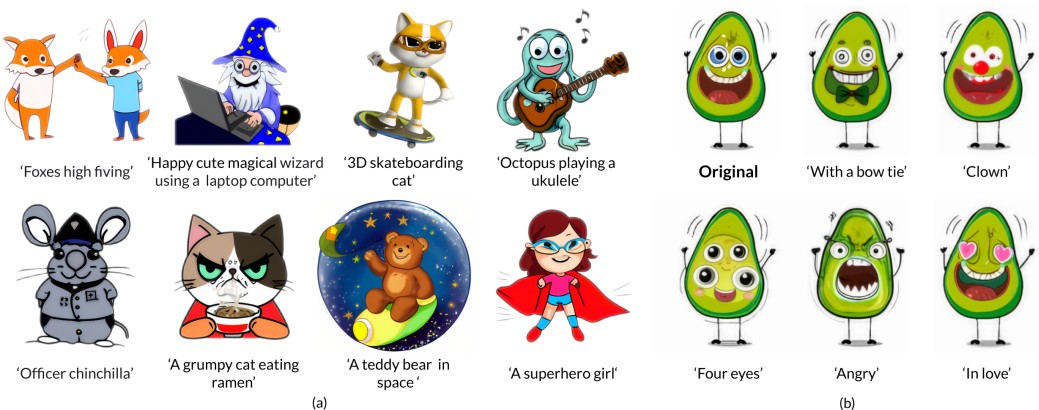

**Figure 1:** (a) Samples of stickers generated from text inputs, (b) Semantic text-guided manipulations applied to the "Original" image without using edit masks.

## Abstract

Recent text-to-image models have achieved impressive results. However, since they require large-scale datasets of text-image pairs, it is impractical to train them on new domains where data is scarce or not labeled. In this work, we propose using large-scale retrieval methods, in particular, efficient $k$-Nearest-Neighbors (kNN), which offers novel capabilities: (1) training a substantially small and efficient text-to-image diffusion model using only pre-trained multi-modal embeddings, but without an explicit text-image dataset, (2) generating out-of-distribution images by simply swapping the retrieval database at inference time, and (3) performing text-driven local semantic manipulations while preserving object identity. To demonstrate the robustness of our method, we apply our kNN approach on two state-of-the-art diffusion backbones, and show results on several different datasets. As evaluated by human studies and automatic metrics, our method achieves state-of-the-art results compared to existing approaches that train text-to-image generation models using images-only dataset.

## 1 Introduction

Large-scale generative models have been applied successfully to image generation tasks (Gafni et al., 2022; Ramesh et al., 2021; Nichol et al., 2021; Saharia et al., 2022; Yu et al., 2022), and have shown outstanding capabilities in extending human creativity using editing and user control. However, these models face several significant challenges: (i) **Large-scale paired data requirement**. To achieve high-quality results, text-to-image models rely heavily on large-scale datasets of (text, image) pairs collected from the internet. Due to the requirement of paired data, these models cannot be applied to new or customized domains with only unannotated images. (ii) **Computational cost and efficiency**. Training these models on highly complex distributions of natural images usually requires scaling the size of the model, data, batch-size, and training time, which makes them challenging to train and less accessible to the community. Recently, several works proposed text-to-image models

trained without an explicit paired text-image datasets. Liu et al. (2021) performed a direct optimization to a pre-trained model based on a CLIP loss (Radford et al., 2021). Such approaches are time-consuming, since they require optimization for each input. Zhou et al. (2021) proposed training with CLIP image embedding perturbed with Gaussian noise. However, to achieve high-quality results, an additional model needs to be trained with an annotated text-image pairs dataset.

In this work, we introduce a novel generative model, *kNN-Diffusion*, which tackles these issues and progresses towards more accessible models for the research community and other users. Our model leverages a large-scale retrieval method, $k$-Nearest-Neighbors (kNN) search, in order to train the model without an explicit text-image dataset. Specifically, our diffusion model is conditioned on two inputs: (1) image embedding (at training time) or text embedding (at inference), extracted using pre-trained CLIP encoder, and (2) kNN embeddings, representing the $k$ most similar images in the CLIP latent space. During training, we assume that no paired text is available, hence condition only on CLIP image embedding and on $k$ additional image embeddings, selected using the retrieval model. At inference, only text inputs are given, so instead of image embeddings, we use the text embedding that shares a joint embedding space with the image embeddings. Here, the kNN image embeddings are retrieved using the *text* embeddings.

The additional kNN embeddings have three main benefits: (1) they extend the distribution of conditioning embeddings and ensure the distribution is similar in train and inference, thus helping to bridge the gap between the image and text embedding distributions (see Fig. 5); (2) they teach the model to learn to generate images from a target distribution by using samples from that distribution. This allows generalizing to different distributions at test time and generating out-of-distribution samples; (3) they hold information that does not need to be present in the model, which allows it to be substantially smaller. We demonstrate the effectiveness of our kNN approach in Sec. 4.

To assess the performance of our method, we train our model on two large-scale datasets: the Public Multimodal Dataset (Singh et al., 2021) and an image-only stickers dataset collected from the Internet. We show state-of-the-art zero-shot results on MS-COCO (Lin et al., 2014), LN-COCO (Pont-Tuset et al., 2020) and CUB (Wah et al., 2011). To further demonstrate the advantage of retrieval methods in text-to-image generation, we train two diffusion backbones using our kNN approach: continuous (Ramesh et al., 2022) and discrete (Gu et al., 2021). In both cases we outperform the model trained without kNN. In comparison to alternative methods presented in Sec. 4, we achieve state-of-the-art results in both human evaluations and FID score, with only 400 million parameters and 7 seconds inference time.

Lastly, we introduce a new approach for local and semantic manipulations that is based on CLIP and kNN, without relying on user-provided masks. Specifically, we fine-tune our model to perform local and complex modifications that satisfies a given target text prompt. For example, given the teddy bear's image in Fig. 4, and the target text "holds a heart", our method automatically locates the local region that should be modified and synthesizes a high-resolution manipulated image in which (1) the teddy bear's identity is accurately preserved and (2) the manipulation is aligned with the target text. We demonstrate our qualitative advantage by comparing our results with two state-of-the-art models, Text2Live (Bar-Tal et al., 2022) and Textual Inversion (Gal et al., 2022), that perform image manipulations without masks (Fig. 4, 21 and 22).

We summarize the contributions of this paper as follows: (1) We propose *kNN-Diffusion*, a novel and efficient model that utilizes a large-scale retrieval method for training a text-to-image model with only pre-trained multi-modal embeddings, but without an explicit text-image dataset. (2) We demonstrate efficient out-of-distribution generation, which is achieved by substituting retrieval databases. (3) We present a new approach for local and semantic image manipulation, without utilizing masks. (4) We evaluate our method on two diffusion backbones, discrete and continuous, as well as on several datasets, and present state-of-the-art results compared to baselines.

## 2 RELATED WORK

**Text-to-image models.** Text-to-image generation is a well-studied task that focuses on generating images from text descriptions. While GANs (Xu et al., 2018; Zhu et al., 2019; Zhang et al., 2021) and Transformer-based methods (Ramesh et al., 2021; Gafni et al., 2022; Yu et al., 2022; Ding et al., 2021) have shown remarkable results, recently impressive results have been attained with dis-

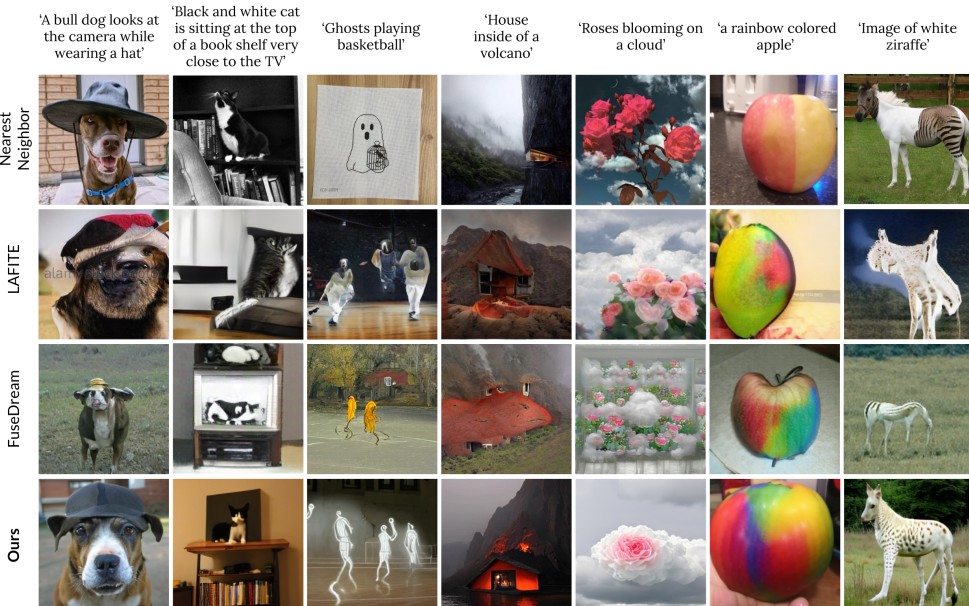

**Figure 2:** Qualitative comparisons with baselines. *Nearest Neighbor* is the first kNN of the text in PMD dataset.

crete (Gu et al., 2021) and continuous (Nichol et al., 2021; Saharia et al., 2022; Ramesh et al., 2022; Rombach et al., 2022) diffusion models. Most recent works trained diffusion models conditioned on text embeddings extracted using a pre-trained text encoder (Saharia et al., 2022; Yu et al., 2022) or image embedding extracted using CLIP (Ramesh et al., 2022). While producing impressive results, all previous works described above are supervised and trained with paired text-image datasets. Several works have proposed training text-to-image models without an explicit text-image dataset. FuseDream (Liu et al., 2021) proposed a direct optimization to a pre-trained generative model based on CLIP loss. This method relies on a pre-trained GAN and requires a time-consuming optimization process for each image. LAFITE (Zhou et al., 2021) recently demonstrated text-to-image generation results without requiring paired text-image datasets. Here, the CLIP embeddings are used interchangeably at train and test to condition a GAN-based model. The joint text-image embedding enables inference given a text input, whereas in training the model is fed with the visual embedding only. However, the gap between the text and image distributions in the joint embeddings space leads to results with substantially lower quality, as we show in our experiments. To overcome this gap, LAFITE added noise to the image embeddings during training. Our remedy to this gap is to condition the model on the retrieval of an actual image embeddings, using a text-image joint space.

**Retrieval for generation.** The Information Retrieval (IR) literature tackles the challenge of retrieving a small amount of information from a large database, given a user's query. A simple, yet efficient retrieval mechanism is to retrieve the $K$ nearest neighbors (kNN) between the query and the entities in the database in some pre-calculated embedding space (Bijalwan et al., 2014). The database allows the model to leverage extensive world-knowledge for its specific task Borgeaud et al. (2021). Recently, language models were augmented with a memory component, allowing them to store representations of past inputs (Wu et al., 2022). The latter were then queried using a lookup operation, improving performance in various benchmarks and tasks. Retrieval models have been used for various tasks in learning problems, for example, language modeling (Borgeaud et al., 2021), machine translation (Gu et al., 2018), question answering (Lee et al., 2019) and image generation (Tseng et al., 2020; Qi et al., 2018). RetrieveGAN (Tseng et al., 2020) uses a differentiable retrieval module for image generation from a scene description, RetrievalFuse (Siddiqui et al., 2021) proposed a neural 3D scene reconstruction based on a retrieval system. SIMS (Qi et al., 2018) proposed generating an image using semantic layout and compatible image segments that are retrieved from image segments database, and (Iskakov, 2018) showed that the use of retrieval database in inpainting task significantly boosts visual quality. In this work we utilize the kNN retrieval mechanism over the shared text-image embedding space, CLIP (Radford et al., 2021). Using extensive ablation studies, we show the importance of the retrieval model both for training and inference, and demonstrate its large impact on performance. *kNN-Diffusion* significantly outperforms prior work

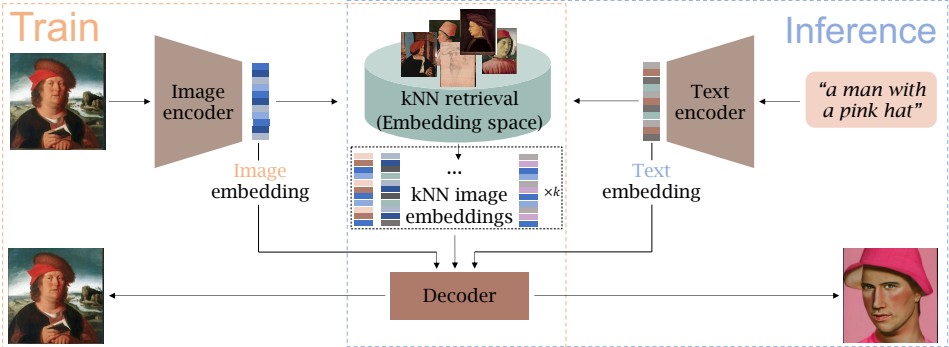

**Figure 3: The overall framework of our *kNN-Diffusion* model.** In both training and inference, the decoder is conditioned on CLIP embedding, and kNN image embeddings. During training, we condition the model on image CLIP embedding, and its kNN image embeddings extracted using the retrieval method. At inference time, given an input text, the kNN image embeddings are retrieved based on the CLIP text embedding that shares a joint embedding space with the image embedding.

with zero-shot FID of 12.5, including RDM (Blattmann et al., 2022)(with FID of 22.1), a concurrent work which similarly to our approach, proposes conditioning LDM (Rombach et al., 2022) on kNN.

**Multi-modal feature learning.** Learning a joint and aligned feature space for several modalities is challenging, as it requires alignment between the modalities (paired datasets), whose distributions may vary. Specifically, the joint feature space of vision-and-language has been a long-standing problem. CLIP (Radford et al., 2021) successfully tackled this by leveraging contrastive learning over a large dataset of text-image pairs. BLIP (Li et al., 2022), (Mu et al., 2021) and FLAVA (Singh et al., 2021), followed this idea and further improved the joint representation. The joint representation was shown to hold a strong semantic alignment between the two modalities, enabling image generation (Liu et al., 2021; Wang et al., 2022), image manipulation (Patashnik et al., 2021; Avrahami et al., 2022b), and image captioning (Mokady et al., 2021). In this work we leverage the joint representation in two ways: (i) enabling textless training with only visual data, while using text at inference time, and (ii) creating an efficient embedding space for the use of the retrieval model.

## 3 METHOD

Our main goal is to facilitate language-guided generation of user-specified concepts while using an images-only dataset during training. A possible way to achieve this goal is to use a shared text-image encoder that will map text-image pairs into the same latent space, thus allowing training with an image embedding, and inferring from text embedding. A candidate for this encoder is CLIP, which has been trained with a contrastive loss on a large-scale dataset of text-image pairs. However, as we show quantitatively in Tab. 1, 2 and qualitatively in Fig. 15, 16, 5, CLIP embeddings alone cannot accurately bridge the gap between the text and image distributions. In order to reduce this gap, several methods have been proposed. The closest work to ours is LAFITE, which perturbs the CLIP image embedding with adaptive Gaussian noise. Under the assumption that there is a large paired text-image dataset, Ramesh et al. (2022) have proposed a prior that is used during inference, and is trained to generate possible CLIP image embeddings from a given text caption. In this regard, we propose using a large-scale and non-trainable image embedding index as an integral part of the diffusion process. Our method, *kNN-Diffusion*, assumes that only image data and a pre-trained multi-modal text-image encoder are provided during training. As shown in Fig. 3, our model is comprised of three main components: (1) A multi-modal text-image encoder (CLIP); (2) A retrieval model - A data structure containing image embeddings, which is indexed for a fast kNN search; (3) An image generation network - A trainable diffusion-based image generation model, conditioned on the projected retrievals. For both training and inference, the image generation network is conditioned on $K$ additional image embeddings, chosen using the retrieval model to ensure a similar distribution of the condition in training and inference. The following sections describe these components.

**Retrieval model.** Our retrieval model has three non-trainable modules: a pre-trained text encoder $f_{txt}$ (CLIP text encoder), a pre-trained image encoder $f_{img}$ (CLIP image encoder) and

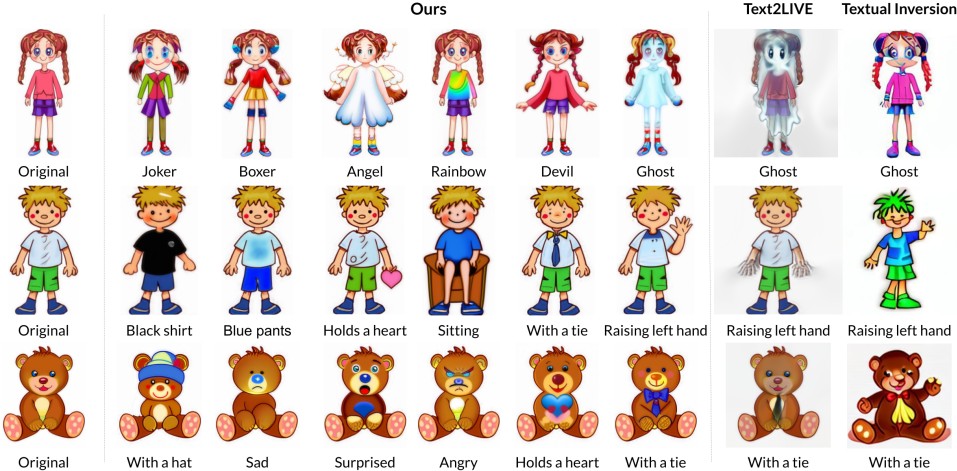

**Figure 4:** Results for text-guided image manipulations without using masks. The original image is shown in the left column, our manipulated images are shown in the center. The images of Bar-Tal et al. (2022); Gal et al. (2022) were generated using the authors' official code. The full comparison is available in the supplement.

an index $\mathcal{H}$. The encoders map text descriptions and image samples to a joint multi-modal $d$-dimensional feature space $\mathbb{R}^d$. The index stores an efficient representation of the images database - $\mathcal{H} := \{f_{img}(i) \in \mathbb{R}^d | i \in \mathcal{I}\}$ where $\mathcal{I}$ denotes the dataset of images. During training, we use the index to efficiently extract the $k$ nearest neighbors in the feature space of the image embedding $f_{img}(\mathbf{I}) \in \mathbb{R}^d$ - $\text{knn}_{img}(\mathbf{I}, k) := \arg\min_{h \in \mathcal{H}}^k \mathbf{s}(f_{img}(\mathbf{I}), h)$ where $\mathbf{s}$ is a distance function and $\arg\min^k$ output the minimal $k$ elements. The set $\{f_{img}(\mathbf{I}), \text{knn}_{img}(\mathbf{I}, k)\}$ is used as the condition to the generative model. During inference, given a query text $t$, an embedding $f_{txt}(t)$ is extracted. The generative model is conditioned on this embedding and its $k$ nearest neighbors from the database - $\text{knn}_{txt}(t, k) := \arg\min_{h \in \mathcal{H}}^k \mathbf{s}(f_{txt}(t), h)$. During training, we add embeddings of real images, by applying the retrieval method to the input image embedding. The extracted kNN should have a large enough distribution to cover the potential text embedding. During inference, the kNN are retrieved using the text embedding (See Fig. 17). In all of our experiments we use the cosine similarity metric as the distance function $\mathbf{s}$, $k = 10$ for the number of nearest neighbors and $d = 512$. The full implementation details can be found in Sec. 6.6 in the supplement.

**Image generation network.** In order to demonstrate the robustness of our method, we apply our kNN approach on two different diffusion backbones: Discrete (Gu et al., 2021) and Continuous (Nichol et al., 2021; Sohl-Dickstein et al., 2015; Ho et al., 2020; Dhariwal & Nichol, 2021). Although very different in practice, these models share the same theoretical idea. Let $x_0 \sim q(x_0)$ be a sample from our images distribution. A forward diffusion process is a Markov chain that adds noise at each step $q(x_n | x_{n-1})$. The reverse process, $p_\theta(x_{n-1} | x_n, x_0)$, is a denoising process that removes noise from an initialized noise state. At inference time, the model can generate an output, starting with noise and gradually removing it using $p_\theta$. For additional background on diffusion models please refer to Sec. 6.1 in the supplement.

In the discrete diffusion model, $q(x_n | x_{n-1}) := v^T(x_n)\mathbf{Q}_n v(x_{n-1})$ where $v(x_n)$ is a one-hot vector with entry 1 at $x_n$, and $\mathbf{Q}_n$ is a transition matrix, modeling the probability to move from state $x_{n-1}$ to $x_n$, using uniform probability over the vocabulary and a pre-defined probability for additional special *[MASK]* token. We can compute the reverse transition distribution according to: $p_\theta(x_{n-1} | x_n, y) := \sum_{\hat{x}_0=1}^{k} q(x_{n-1} | x_n, \hat{x}_0) p_\theta(\hat{x}_0 | x_n, x_0, y)$ where $x_0$ is a discrete vector, tokenized by the VQGAN (Esser et al., 2021) encoder and $y$ is the conditioning signal. For modeling $p_\theta$ we have followed (Gu et al., 2021) and used a conditional Transformer (Vaswani et al., 2017).

In the continuous diffusion model, $q(x_n | x_{n-1}) := \mathcal{N}(x_n; \sqrt{\alpha_t}x_{n-1}, (1 - \alpha_n)x_0)$ and $p_\theta(x_{n-1} | x_n, y) := \mathcal{N}(\mu_\theta(x_n, y), \Sigma_\theta(x_n, y))$. Here, the noise function is Gaussian noise. Following (Ho et al., 2020; Nichol et al., 2021) we trained a model $\epsilon_\theta$ to predict the added noise using a standard mean-squared error loss: $L := E_{n \sim [1,N], x_0 \sim q(x_0), \epsilon \sim \mathcal{N}(0,\mathbf{I})}[||\epsilon - \epsilon_\theta(x_n, n, y)||^2]$ where $\epsilon_\theta$ is a U-net model and $y$ is the conditioning signal.

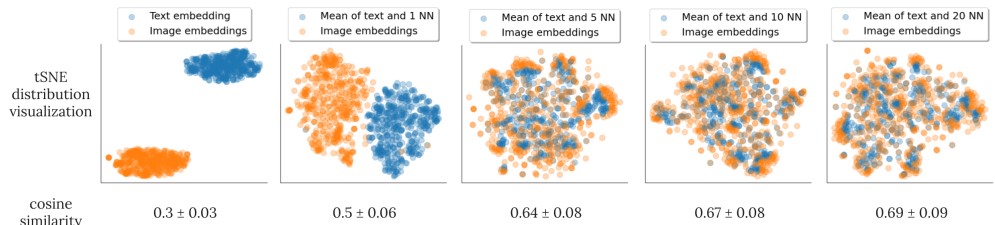

**Figure 5:** *tSNE* visualization of 500 random text-image CLIP embeddings pairs taken from COCO validation. The leftmost figure demonstrates the gap between the text and image distributions. By gradually adding kNN to the mean CLIP embedding of the text, the gap decreases, demonstrating the importance of the kNN.

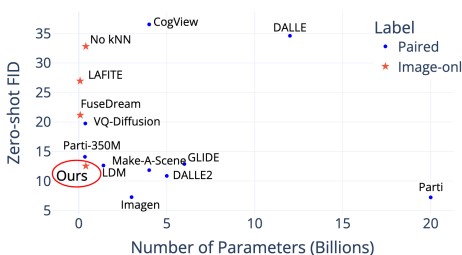

**Figure 6:** FID on MS-COCO, including models trained on image-only datasets and text-image datasets.

**Table 1:** Results for zero-shot Text-to-Image generation on the MS-COCO, CUB and LN-COCO test sets. Image-quality and Text-alignment report the percentage of majority human raters votes in favor of our method when comparing between a certain model and ours.

| Model | MS-COCO FID↓ | Im. qual. | Txt align. | CUB FID↓ | Im. qual. | Txt align. | LN-COCO FID↓ | Im. qual. | Txt align. |
|---|---|---|---|---|---|---|---|---|---|
| LAFITE | 26.9 | 72.1 | 65.3 | 89.7 | 74.0 | 59.6 | 42.8 | 68.4 | 61.9 |
| FuseDream | 21.2 | 64.0 | 79.3 | 50.2 | 79.1 | 60.9 | 37.5 | 71.1 | 59.0 |
| *no-kNN* | 32.8 | 70.8 | 68.3 | 95.1 | 81.0 | 61.2 | 65.0 | 61.4 | 59.8 |
| Ours | **12.5** | - | - | **42.9** | - | - | **35.6** | - | - |

In both cases, we condition our model on $y = (f_{img}(x_0), \mathrm{knn}_{img}(x_0, k))$ where $f_{img}(x_0)$ is the CLIP image embedding, $\mathrm{knn}_{img}(x_0, k)$ is the $k$ nearest neighbors in the feature space of the image embedding. Following (Ramesh et al., 2022; Rombach et al., 2022) conditional injection, we condition our model on the image CLIP embedding, and the kNN clip embeddings by applying cross attention in the attention layers of the architecture. We sample both our models using Classifier Free Guidance (CFG) (Nichol et al., 2021; Ho & Salimans, 2021). Since CFG was originally proposed for continuous models, we propose a method for using it with discrete models as well. Full implementation details of the discrete and continuous models can be found in Sec. 6.7 and Sec. 6.8, respectively, in the supplement.

## 3.1 TEXT-ONLY IMAGE MANIPULATION

The majority of previous works in the task of image manipulation either rely on user-provided masks (Nichol et al., 2021; Avrahami et al., 2022b;a), or are limited to global editing (Crowson et al., 2022; Kim et al., 2022). Recently, several works (Bar-Tal et al., 2022; Hertz et al., 2022; Gal et al., 2022) have made progress with local manipulations without relying on user edited masks. Nevertheless, most of the techniques suffer from several shortcomings: (1) They enable local texture changes, yet cannot modify complex structures, (2) they struggle to preserve the identity of the object, for example, when manipulating humans, (3) they require optimization for each input.

We address these issues by extending *kNN-Diffusion* to perform local and semantic-aware image manipulations without any provided mask. Illustration of the approach is provided in Fig. 18 and Fig. 19 in the supplement. For this task, the model is trained to predict the original image from a manipulated version. Specifically, we create a manipulated version of the image, which differs from the original image only in some local area. Given a random local area $M$ in the image I, the manipulated image $\mathrm{I}_{manip}$ is constructed by replacing the area with the corresponding nearest neighbor: $\mathrm{I}_{manip} = \mathrm{I} \cdot (1 - M) + \mathrm{nn}_{img}(\mathrm{I}, 1) \cdot M$, where $\mathrm{nn}_{img}(\mathrm{I}, 1)$ is the the nearest neighbor obtained after aligning it with I using the ECC alignment algorithm (Evangelidis & Psarakis, 2008). The model then receives as input the manipulated image, together with the CLIP embedding of the original image *only* in the local area: $f_{img}(\mathrm{I} \cdot M)$. This CLIP embedding represents the required modification that should be applied to the manipulated image in order to predict the original image. During inference, instead of using the CLIP embedding of the local area, the desired modification is

**Table 2:** Results on the stickers dataset. We report the percentage of human raters prefer our method over the baselines with respect to image quality and text alignment. Discrete *no-kNN* refers to VQ-diffusion, and Continuous *no-kNN*, to DALL·E2 decoder, both trained without an explicit text-image dataset.

| Model | FID↓ | Ours Discrete | | Ours Continuous | |
|---|---|---|---|---|---|
| | | Image quality | Text alignment | Image quality | Text alignment |
| DALL·E2+ClipCap | 55.5 | 71.6 | 69.2 | 67.0 | 68.3 |
| LAFITE | 58.7 | 63.5 | 59.9 | 76.0 | 71.2 |
| *no-kNN* | 52.7 | 72.1 | 67.6 | 66.8 | 69.4 |
| Ours | **40.8** | - | - | - | - |

represented using the CLIP embedding of the user text query. We modified the model to be capable of receiving as a condition both the manipulated image and the CLIP embedding of the local area.

# 4 Experiments

First, we conduct qualitative and quantitative comparisons on MS-COCO, LN-COCO and CUB datasets. To further demonstrate the advantage of our method, we provide comparison on an image-only stickers dataset, where we apply our approach on two diffusion backbones. Next, we demonstrate image manipulation and out-of-distribution capabilities. Finally, to better assess the effect of each contribution, an ablation study is provided.

**Datasets and Metrics.** For photo-realistic experiments, our model was trained only on the images (omitting the text) of a modified version of the Public Multimodal Dataset (PMD) used by FLAVA (Singh et al., 2021). More information about the dataset is available in Sec. 6.4 of the supplement. To further demonstrate the capabilities of our method, we collected 400 million sticker images from the web, containing combinations of concepts such as objects, characters/avatars and text. The collected stickers do not have paired text, and are substantially different from photo-realistic data. Furthermore, since they have no paired text, they were not part of CLIP's training data, which makes the text-to-image generation task more challenging.

Evaluation metrics are based on objective and subjective metrics: (i) FID (Heusel et al., 2017) is an objective metric used to assess the quality of synthesized images, (ii) human evaluation - we ask human raters for their preference, comparing two methods based on image quality and text alignment. We used 600 image pairs; five raters rated each pair. The results are shown as a percentage of majority votes in favor of our method over the baselines. We report the full human evaluation protocol in the supplement. We chose to omit Inception-Score, since it is shown by Barratt & Sharma (2018) to be a misleading metric for models that were not trained on Imagenet.

## 4.1 Qualitative and Quantitative Results

We begin by comparing our model, trained on the PMD dataset, with the previous works LAFITE and FuseDream, that trained on image-only datasets. To demonstrate the advantage of using a retrieval method in text-to-image generation, we trained a model variant, *no-kNN*. This baseline was trained solely on image embeddings (omitting the kNN), while during inference, the images were generated using the text embedding. Tab. 1 displays zero-shot results on three different datasets: MS-COCO, CUB and LN-COCO. We follow the evaluation protocol of LAFITE, reporting our results on 30,000 images from MS-COCO validation set without training, nor using it's training partition in the kNN index. Similarly, we follow LAFITE for CUB and LN-COCO evaluation. As can be seen, our model achieves the lowest FID score in all scenarios. In addition, human evaluations rate our method as better aligned to text and with the highest images quality. In Fig. 2, 15 and 11 we present a qualitative comparison between the methods. One can observe that while the simple retrieval baseline outputs non-generated images with high-quality, the images generated by our method are more faithful to the input text. To further demonstrate the effectiveness of our method, we present in Fig. 6 a comparison of our model with the latest text-to-image models trained on paired text-image datasets: DALL·E, CogView, VQ-Diffusion, GLIDE, LDM, Make-A-Scene, DALL·E2, Parti and Imagen. As can be seen, our model achieves comparable results to recent models trained with full text-image pairs (e.g LDM, GLIDE), despite being trained on an image-only dataset, with significantly lower computational costs. The results demonstrate that leveraging an external retrieval database allows to compensate for different trade-offs, in particular, reducing the number of parameters in the model. Additional samples are provided in Fig. 13 in the supplement.

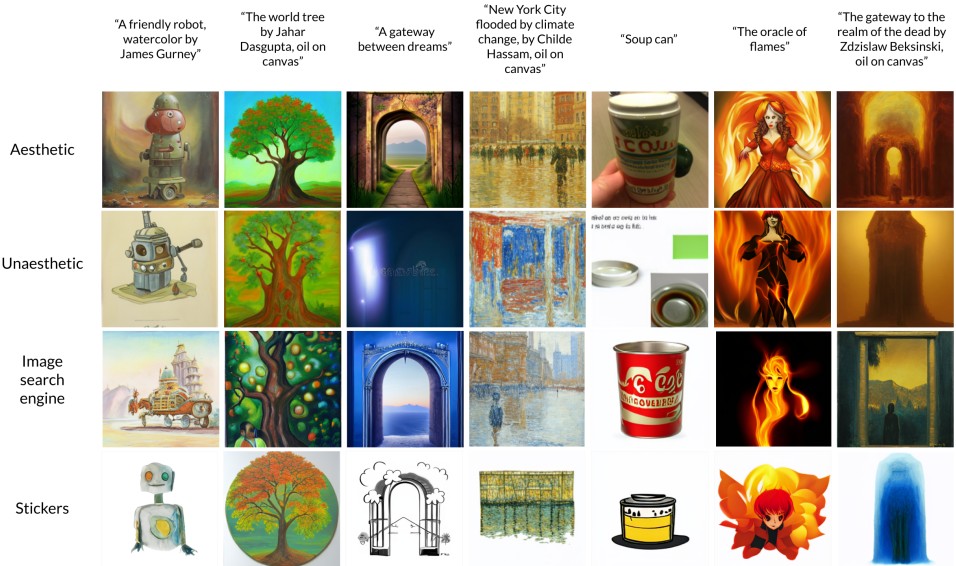

**Figure 7:** Comparison between various indexes used by the same model. (1) **Aesthetic.** Images from the first quantile of an aesthetic classifier, (2) **Unaesthetic.** Images from the last quantile of an aesthetic classifier, (3) **Image search engine.** Images retrieved from Google Images, (4) **The stickers index.**

**Text-to-sticker generation.**  As the sticker dataset does not have paired text, and is substantially different from photo-realistic data, it allows us to illustrate the advantage of our model on an image-only dataset. A selection of stickers generated by our model is presented in Fig. 1 and Fig. 14, 12. To demonstrate the importance of using kNN on image-only datasets, we evaluate our approach on two diffusion backbones. To this end, we trained a continuous diffusion model  (Ramesh et al., 2022) and a discrete diffusion model  (Gu et al., 2021), both conditioned on the kNN image embeddings. For each backbone, we compare our method with the following baselines: (1) *no-kNN* - this baseline was trained using both the continuous and the discrete methods conditioned only on image CLIP embedding, without using kNN. In the discrete case, we trained a VQ-diffusion model, while in the continuous case, we trained a re-implementation of DALL·E2's decoder (without prior). (2) *DALL·E2+ClipCap* baseline - here, we first captioned the entire sticker dataset using Clip-Cap (Mokady et al., 2021), then trained DALL·E2 decoder on the captioned dataset. (3) LAFITE - we trained LAFITE language-free model on our stickers dataset using the authors' published code. We present the results in Tab. 2. The FID is calculated over a subset of 3,000 stickers, generated from the ClipCap captioned dataset. As can be seen, our model achieves the lowest FID score. In addition, it outperforms all baselines in human evaluation comparison, using continuous and discrete backbones. In particular, compared with the same model trained without kNN, our model achieves significantly higher favorability in both text alignment and image quality.

## 4.2 APPLICATIONS

**Text-only image manipulation.**  We demonstrate the manipulation capabilities of our model in Fig. 1, 4 and 20. Furthermore, we qualitatively compare our model with Text2LIVE (Bar-Tal et al., 2022) and Textual Inversion (Gal et al., 2022), using the authors' published code. Text2LIVE proposed generating an edit layer that is composed over the original input, using a generator trained for each training image. Textual Inversion utilized the pre-trained Latent Diffusion model to invert the input image into a token embedding. The embedding is then used to compose novel textual queries for the generative model. Fig. 4 shows representative results, and the rest are included in Fig. 21 and 22 in the supplement. In contrast to our model, baseline methods lack text correspondence or they do not preserve the identity of the object. Since Text2LIVE is optimized to perform local changes, it has the difficulty changing the structure of the object (e.g. the "raising his hand" example in Fig. 4). Textual Inversion baseline changes the identity of the object because it struggles reconstructing the textual representation of the source image. Our model, on the other hand, can perform challenging manipulations that are aligned with the text, while preserving the object identity.

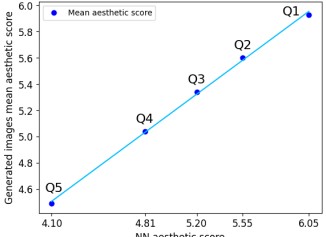 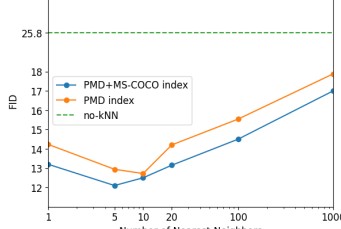 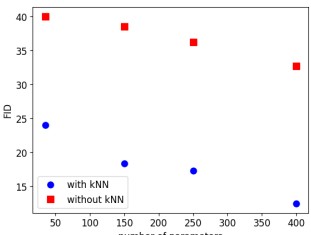

**Figure 8:** Mean aesthetics score of the generated images as a function of the conditioned kNN mean aesthetics score.

**Figure 9:** MS-COCO test FID score on various K's in: (1) Zero-Shot (2) Index includes MS-COCO train subset. **No kNN** trained with kNN, but did not employ kNN in inference.

**Figure 10:** MS-COCO test FID score for different model sizes. As can be seen, adding kNN to the model allows it to be smaller, while having better performance.

**Out-of-distribution generation.** Using the retrieval index as part of the generation process enables using different databases during inference, without fine-tuning. This allows generatig images from distributions that were not part of the training set, enabling out-of-distribution generation. This novel capability is demonstrated with the same model trained on PMD, using three different retrieval databases: *(i) A stickers database* presented in Sec. 4. *(ii) Aesthetic database:* This database is constructed by filtering images according to a classifier score. Let $C$ be a classifier that for each image $i \in I$ outputs a score $s = C(i)$. This classifier enables filtering the kNN using $L \leq s < H$, where $L$ and $H$ are low and high thresholds, respectively. Here, we use an open source pre-trained aesthetics classifier $A$ (Christoph Schuhmann, 2022): For each text input $t \in T$, we apply $A$ on the kNN, and then divide the kNN into five equal quantiles based on $A$ score. As can be seen in Fig. 8, using kNN with higher aesthetics score result in generated images with higher aesthetics mean score. *(iii) Image search engine:* Generative models are stationary in the sense that they are unable to learn new concepts after being trained, hence fine-tuning is required to represent new styles and concepts. Here, we use an online image search engine, which allows the model to adapt to new data without additional fine-tuning. A qualitative comparison of all three methods is shown in Fig.7.

### 4.3 ABLATION STUDY

We conclude our experiments with an ablation study, to quantify the contribution of our different components. We provide ablation study on index size and different kNN conditioning approaches in Sec. 6.5 of the supplement. **Number of nearest neighbors.** The results in Fig. 9 demonstrate the importance of applying the retrieval mechanism during training and inference. Here, we evaluate our model, trained on PMD dataset, with different numbers of kNN during inference. Furthermore, we examined the baseline *no-kNN*, in which during inference, the model is conditioned only on the text embedding $f_{txt}(t)$, without using kNN. Best performance is achieved using 10 neighbors. **Scalability analysis.** To evaluate the effectiveness of our approach at different model sizes, we trained three additional models with varying sizes for both settings - with and without kNN. As can be seen in Fig. 10, utilizing kNN consistently improves performance for all sizes. Furthermore, a performance improvement can be achieved using much smaller models with kNN. For example, the $35M$ kNN model outperforms the $400M$ model without kNN.

### 5 CONCLUSION

"We shall always find, that every idea which we examine is copied from a similar impression", Hume (1748). In this paper, we propose using a large-scale retrieval method in order to train a novel text-to-image model, with only pre-trained multi-modal embeddings, but without an explicit text-image dataset. Our extensive experiments demonstrate that using an external knowledge-base alleviates much of the model's burden of learning novel concepts, enabling the use of a relatively small model. In addition, it provides the model the capability of learning to adapt to new samples, which it only observes during test time. Lastly, we present a new technique utilizing the retrieval method for text-driven semantic manipulations without user-provided masks. As evaluated by human studies and automatic metrics, our method is significantly preferable to the baselines in terms of image quality and text alignment.

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

# 6 APPENDIX

**Figure 11:** Samples from COCO validation set.

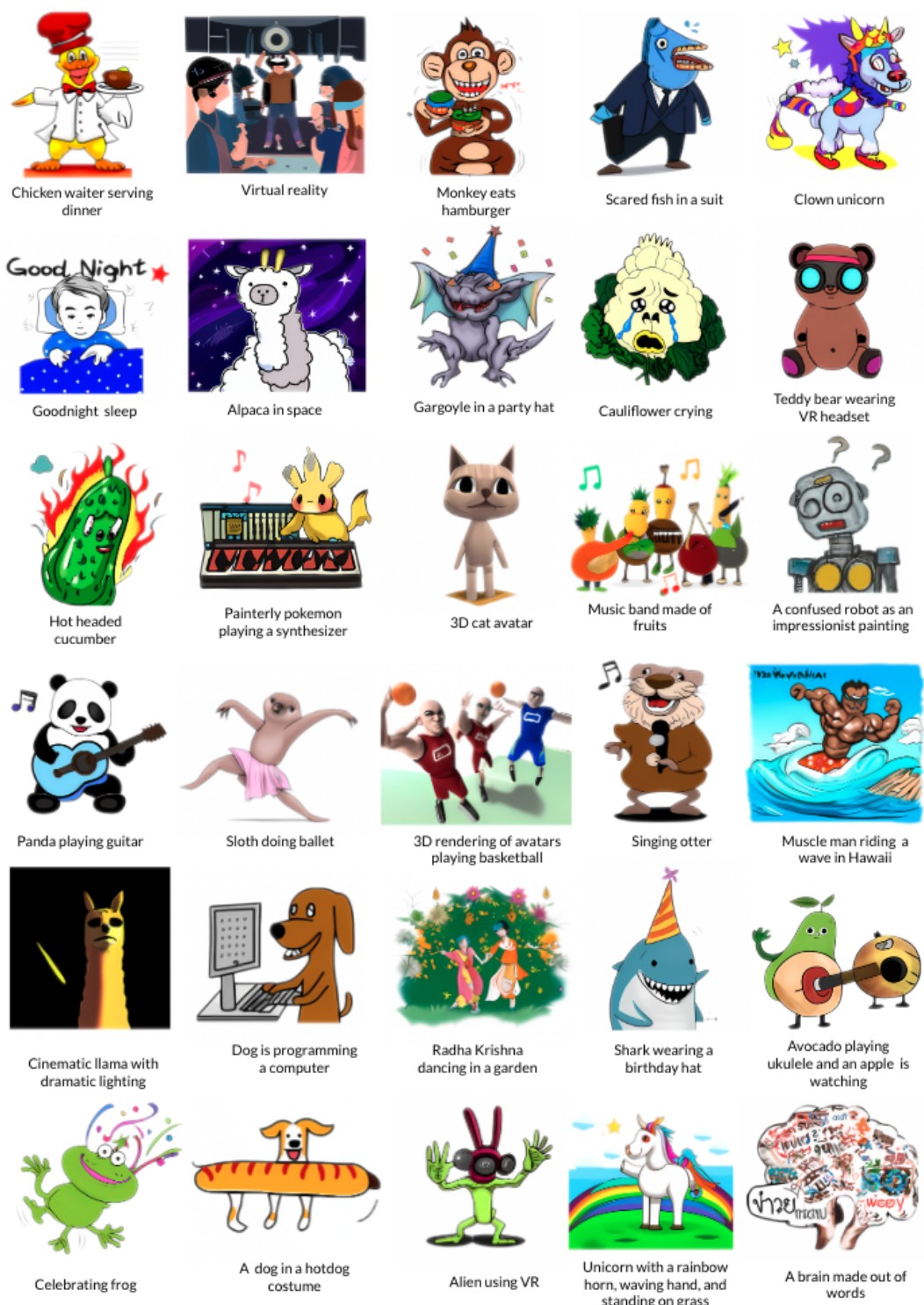

**Figure 12:** A selection of stickers generated using the continuous *kNN-Diffusion* model.

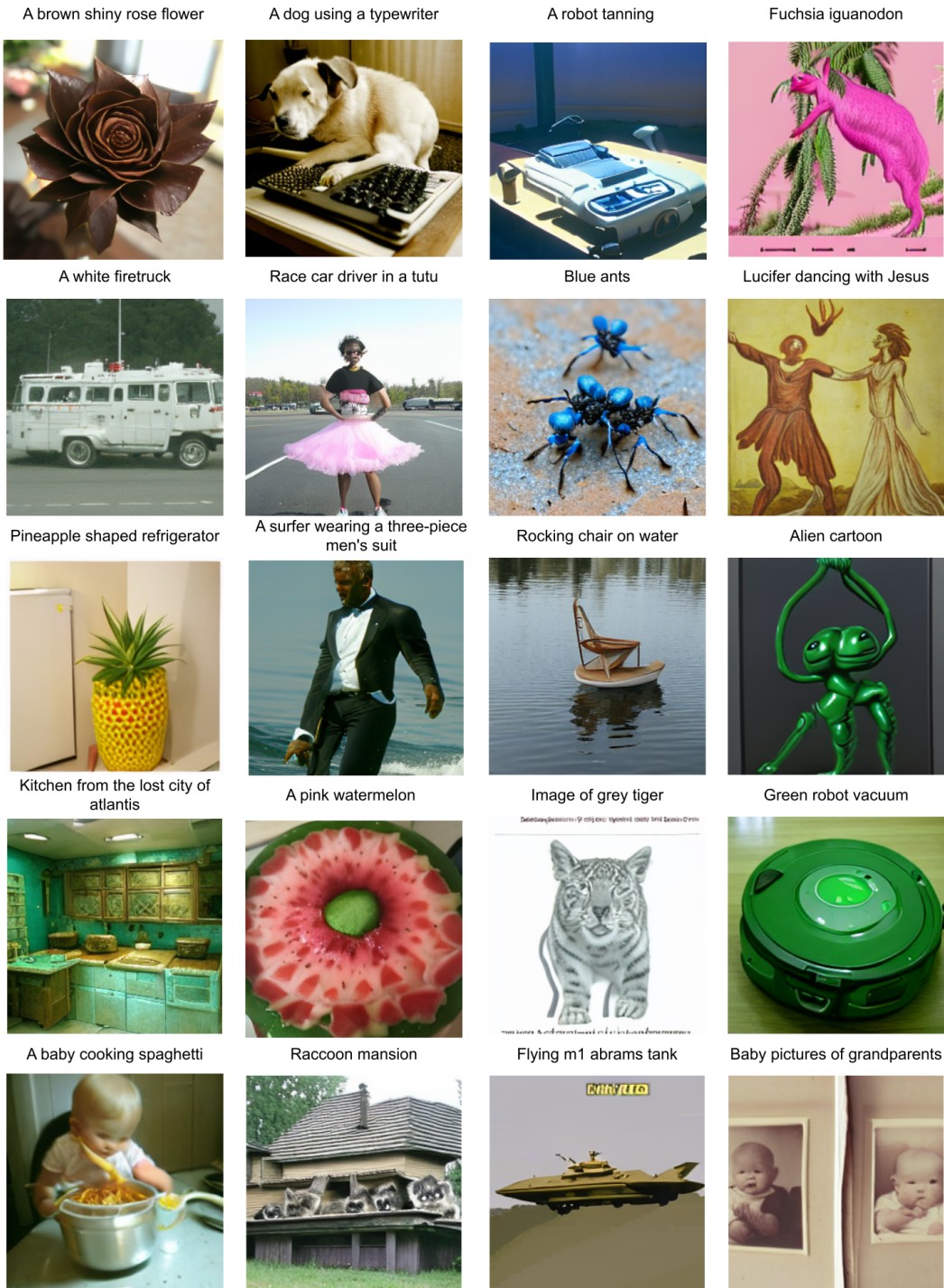

**Figure 13:** Additional samples generated from challenging text inputs using the photo-realistic model

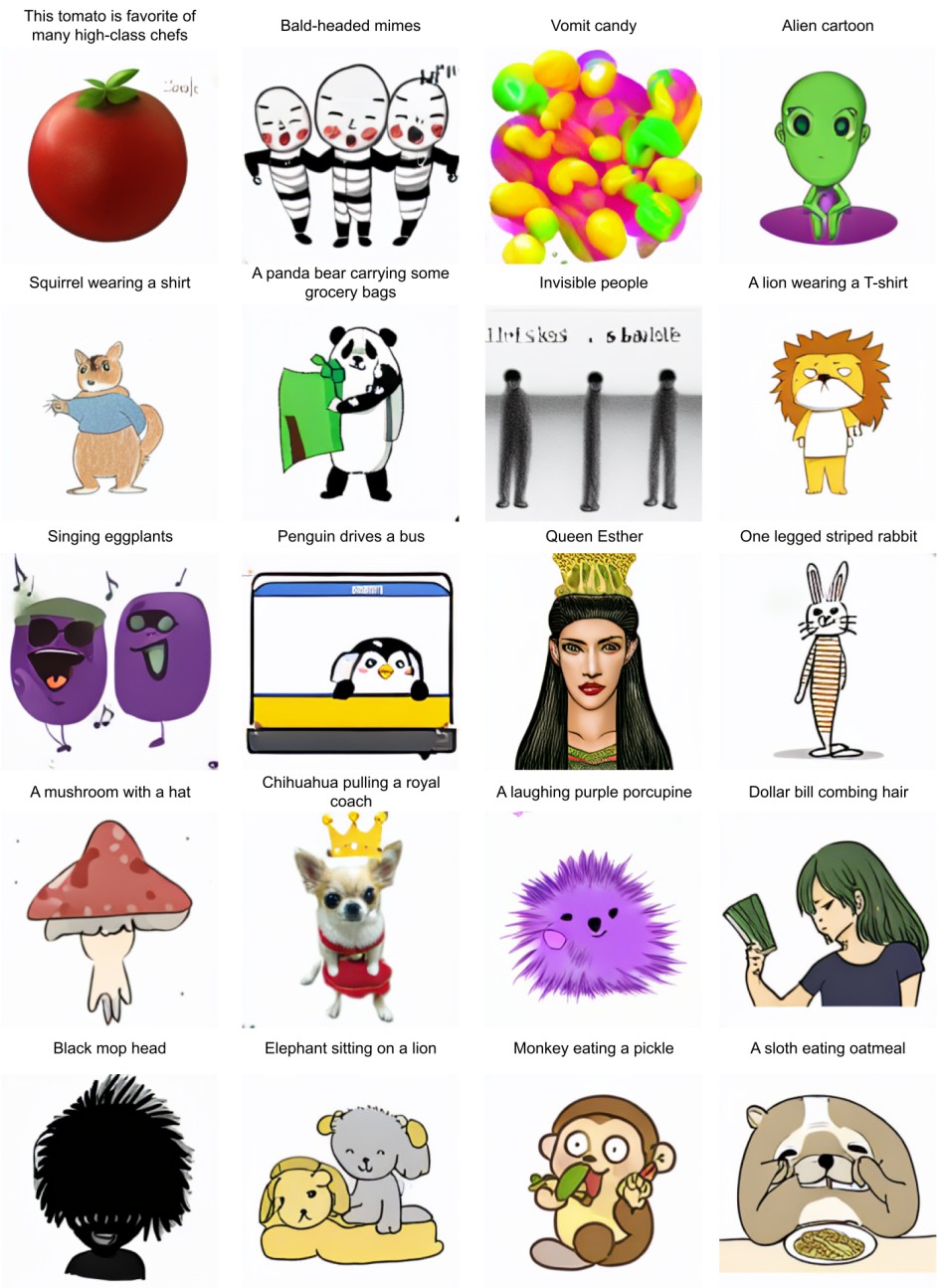

**Figure 14:** A selection of stickers generated using the discrete *kNN-Diffusion* model.

## 6.1  BACKGROUND

**Continuous diffusion process**   Diffusion models are latent variable models that aim to model a distribution $p_\theta(x_0)$ that approximates the data distribution $q(x_0)$. Specifically, they model a forward process in the space of $x_0$ from data to noise. Given a sample from the data distribution $x_0 \sim q(x_0)$, this process produces a Markov chain of latent variables $x_1, \ldots, x_T$ by progressively adding Gaussion noise to the sample:

$$q(x_t|x_{t-1}) := \mathcal{N}(x_t; \sqrt{1 - \beta_t}x_{t-1}, \beta_t\mathcal{I}) \tag{1}$$

where $\beta_t$ is a variance schedule. As presented previously by (Ho et al., 2020), the latent variable $x_t$ can be expressed directly as a linear combination of noise and $x_0$:

$$x_t = \sqrt{\overline{\alpha_t}}x_0 + \epsilon\sqrt{1 - \overline{\alpha_t}}, \quad \epsilon \sim \mathcal{N}(0, \mathcal{I}) \tag{2}$$

where $\alpha_t := \Pi_{i=1}^t(1 - \beta_i)$. In order to sample from the data distribution $q(x_0)$, we define the "reverse process" $p(x_{t-1}|x_t)$ which samples first from $q(x_T)$ and then samples reverse steps $q(x_{t-1}|x_t)$ until $x_0$.

Since the data distribution is unknown, we need to train a model to approximate it. Note that when $T$ is large enough, the noise vector $x_T$ nearly follows an isotropic Gaussian distribution. This suggests learning a model $p_\theta(x_{t-1}|x_t)$ to predict mean $\mu_\theta$ and covariance matrix $\Sigma_\theta$:

$$p_\theta(x_{t-1}|x_t) := \mathcal{N}(x_{t-1}; \mu_\theta(x_t, t), \Sigma_\theta(x_t, t)) \tag{3}$$

To train this model, we can replace $\mu_\theta(x_t, t)$ by predicting the noise $\epsilon_\theta(x_t, t)$ added to $x_0$ using equation 2 and we get this objective function:

$$L := E_{t \sim [1,T], x_0 \sim q(x_0), \epsilon \sim \mathcal{N}(0,\mathbf{I})}[||\epsilon - \epsilon_\theta(x_t, t, y||^2] \tag{4}$$

where $y$ is an optional conditioning signal (such as text/image embedding or a low resolution image).

**Discrete diffusion process**   Let $x_n \in \{1, \ldots, V\}^{h \times w}$ be the indices of the allocated codebook vectors extracted by a pre-trained VQGAN (Esser et al., 2021) encoder. The forward process of a diffusion model $q(x_n|x_{n-1})$ is a Markov chain that adds noise at each step. Moreover, the reverse process $q(x_{n-1}|x_n, x_0)$, is a denoising process that removes noise from an initialized noise state. As presented by (Gu et al., 2021), the forward diffusion process is given by:

$$q(x_n|x_{n-1}) = v^T(x_n)\mathbf{Q}_n v(x_{n-1}) \tag{5}$$

where $v(x_n)$ is a one-hot vector with entry 1 at $x_n$, and $\mathbf{Q}_n$ is the probability transition matrix from state $x_{n-1}$ to $x_n$.

The reverse process is given by the posterior distribution:

$$q(x_{n-1}|x_n, x_0) = \frac{(v^T(x_n)\mathbf{Q}_n v(x_{n-1}))(v^T(x_{n-1})\bar{\mathbf{Q}}_{n-1}v(x_0))}{v^T(x_n)\bar{\mathbf{Q}}_n v(x_0)} \tag{6}$$

where $\bar{\mathbf{Q}}_n = \mathbf{Q}_n \cdots \mathbf{Q}_1$.

Inspired from mask language modeling (Devlin et al., 2018), they proposes corrupting the tokens by stochastically masking some of them. Specifically, an additional special token $[MASK]$ is proposed, so for each token there are (V+1) discrete states. The transition matrix is formulated as, By adding a small amount of unifrom noise to the categorial distribution, the transition matrix can be formulated as,

$$\mathbf{Q}_n = \begin{bmatrix} \alpha_n + \beta_n & \beta_n & \beta_n & \cdots & 0 \\ \beta_n & \alpha_n + \beta_n & \beta_n & \cdots & 0 \\ \beta_n & \beta_n & \alpha_n + \beta_n & \cdots & 0 \\ \vdots & \vdots & \vdots & \ddots & \vdots \\ \gamma_n & \gamma_n & \gamma_n & \cdots & 1 \end{bmatrix} \tag{7}$$

where $\alpha_n \in [0, 1]$, $\beta_n = (1 - \alpha_n - \gamma_n)/V$ and $\gamma_n$ the probability of a token to be replaced with a $[MASK]$ token. Each token has a probability of $\gamma_n$ to be replaced by the $[MASK]$ token, $V\beta_n$ to be resampled uniformly and $\alpha_n = (1 - V\beta_n - \gamma_n)$ to be unchanged.

## 6.2 Additional Samples

In Fig. 16 and 15 we present a visual comparison of our discrete model, trained on the stickers dataset with (1) the kNN extracted during inference, (2) the same model without using kNN in inference. As can be seen, the images generated by our model are better aligned to the corresponding text compared to the baselines. While the baselines fail with challenging prompts, our model produces high-quality images that align with the text, and composes multiple concepts correctly.

**COCO Validation Set Comparison** Fig. 11 presents a qualitative comparison with Fuse-Dream (Liu et al., 2021), CogView (Ding et al., 2021) and VQ-Diffusion (Gu et al., 2021) on the COCO validation set. Note that both CogView and VQ-Diffusion have been trained on an Image-Text *paired* dataset, whereas our model was not trained on the COCO dataset, nor used it in the retrieval model.

Additional samples generated from challenging text inputs are provided in Figs. 13, 14 and Fig. 12.

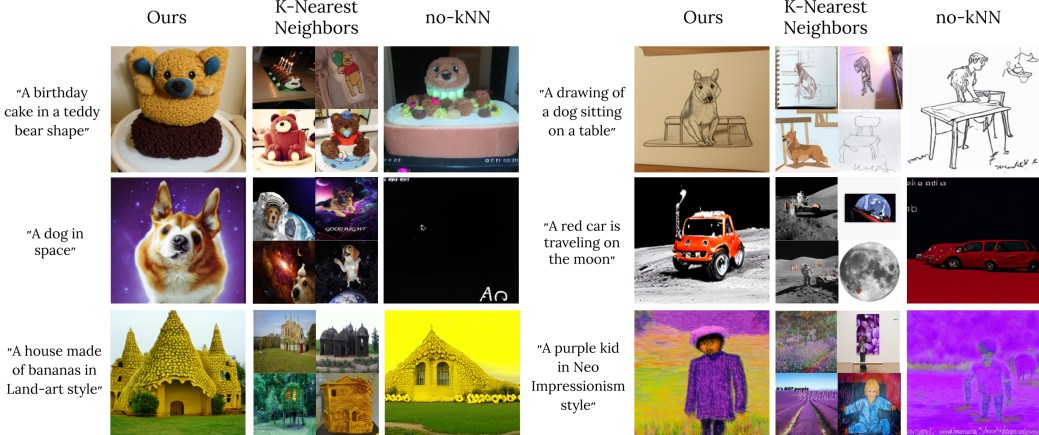

**Figure 15:** Comparison of our model, trained on PMD with (1) kNN extracted in inference, (2) the same model without using kNN in inference. While the kNN lack information regarding text semantics, our model considers both text semantics and the kNN, thus proving the advantage of using both the text and the kNN embeddings.

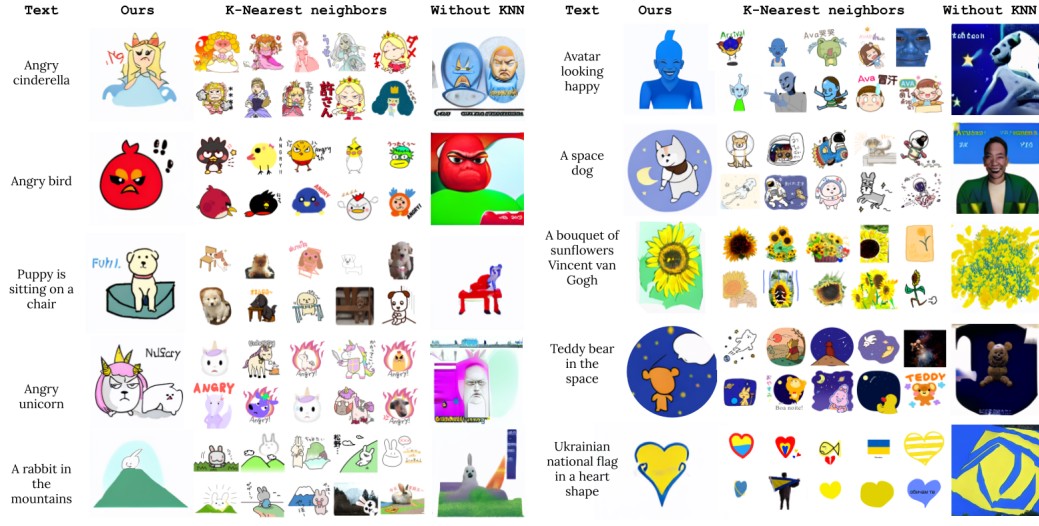

**Figure 16:** Qualitative comparison of stickers generated using the discrete *kNN-Diffusion* model, 10 Nearest Neighbors to the text in the CLIP embedding and a discrete model that does not use kNN.

## 6.3 HUMAN EVALUATION PROTOCOL

For all of our human evaluation experiments, we used Amazon Mechanical Turk. For each experiment, we used 600 samples, each scored by five different people. The preferred sample was determined according to majority opinion. For each baseline comparison, we asked two questions (in different experiments): "Which image is of a higher quality?" and "Which image best matches the text?".

## 6.4 DATASETS

The modified PMD dataset is composed of the following set of publicly available text-image datasets: SBU Captions (Ordonez et al., 2011), Localized Narratives (Pont-Tuset et al., 2020), Conceptual Captions (Sharma et al., 2018), Visual Genome (Krishna et al., 2016), Wikipedia Image Text (Srinivasan et al., 2021), Conceptual Captions 12M (Changpinyo et al., 2021), Red Caps (Desai et al., 2021), and a filtered version of YFCC100M (Thomee et al., 2015). In total, the dataset contains 69 million text-image pairs.

## 6.5 ABLATION STUDY

**Index size** As one can expect, increasing the index size at inference time improves performance. To demonstrate this hypothesis, we evaluated our model with an index containing 10%, 30%, 50% and 70% images of PMD dataset, and obtained FID scores of 13.92, 13.85, 13.72, and 13.65 respectively.

**kNN conditioning** We examined several different approaches to kNN input conditioning: (i) forwarding the kNN embeddings and the single image embedding through a self-attention layer before feeding the contextualized $K + 1$ embeddings to the model, (ii) feeding the model with one embedding, computed using cross-attention between the image embedding and the kNN embeddings, and, (iii) feeding the model with the image embedding concatenated with a learned linear projection of the kNN embeddings. These variants received FID scores of 18.3, 22.4, 34.1 respectively.

## 6.6 RETRIEVAL MODEL

The retrieval model is implemented using FAISS (Johnson et al., 2019). FAISS is an efficient database, capable of storing billions of elements and finding their nearest neighbors in milliseconds. In the pre-processing phase, for each image in the dataset, we store the image index and its corresponding CLIP image embedding. During training, given a training image, we extract its CLIP image embedding and search for its 10 (see Fig. 9) nearest neighbors in the dataset based on the cosine similarity distance.

For an efficient search during training and inference, we use a non-exhaustive search: For this, we use an inverted file index. As in Babenko & Lempitsky (2014), we define Voronoi cells in the $d$-dimensional space (where $d = 512$ is the CLIP embedding dimensional space), s.t each database vector falls in one of the cells. During search time, only the embeddings contained in the cell the query falls in and a few neighboring ones are compared against the query vector. In addition, to fit the index of our large-scale datasets on a 128GB RAM server, we compress the code size from $512 \times 32/8 = 2048$ Bytes to 256 Bytes using optimized product quantization (Ge et al., 2013; Jegou et al., 2010). In Algorithm 1 we include pseudocode of the core of the implementation of the retrieval database.

## 6.7 DISCRETE KNN MODEL

We provide additional implementation details for the discrete diffusion model. Additional training details can be found in Tab. 3.

**Vector Quantization** For token quantization, we use VQ-VAE and adapt the publicly available VQGAN(Esser et al., 2021) model, trained on the OpenImages(Krasin et al., 2016) dataset. The encoder downsamples images to $32 \times 32$ tokens and uses a codebook vocabulary with 2887 elements.

**Image Tokenization** In our discrete generative model we model images as a sequence of discrete tokens. To this end, we utilize a vector-quantized variational auto-encoder (VQ-VAE) (Van Den Oord et al., 2017) as image tokenizer. VQ-VAE consists of three components: (i) an encoder, (ii) a learned codebook, and, (iii) a decoder. Given an image, the encoder extracts a latent representation. The codebook then maps each latent vector representation to its nearest vector in the codebook. Finally, the decoder reconstructs the image from the codebook representation. VQ-VAE is trained with the objectives of reconstruction and codebook learning. VQ-GAN (Esser et al., 2021) adds an adversarial loss term that tries to determine whether the generated image is fake or real. This added term was shown to improve reconstruction quality.

**Transformer** We follow Gu et al. (2021) and train a decoder-only Transformer. The decoder module contains 24 transformer blocks, each containing full attention, cross-attention for the concatenated conditioner, and a feed-forward network. The timestamp $n$ is injected using Adaptive Layer Normalization (Ba et al., 2016). The decoder contains 400 million parameters.

**Classifier-free guidance** We sample our diffusion models using classifier-free guidance (CFG) (Ho & Salimans, 2021; Nichol et al., 2021; Ramesh et al., 2022). CFG is performed by extrapolating an unconditional sample in the direction of a conditional sample. To support unconditional sampling, previous work had to fine-tune (Nichol et al., 2021) their models with 20% of the conditional features nullified. This enabled them to sample unconditional images from the model using the null condition, $y' = \overrightarrow{0}$, the null vector. We found that we can generate unconditional samples from our model using null conditioning without fine-tuning it. We hypothesize that by conditioning the model on a null vector, the cross-attention component is also nullified, resulting in no contribution to the diffusion process. During inference, in each step of the diffusion process we generate two images: conditional image logits, $p_\theta(x_{n-1}|x_n, y)$, conditioned on the desired multi-modal embedding $y$, and the unconditional image logits, $p_\theta(x_{n-1}|x_n, y')$, conditioned on the null embedding. Then, the final image for a diffusion step $n$ is sampled from

$$p_\theta(x_{n-1}|x_n, y) = p_\theta(x_{n-1}|x_n, y') + \\ \lambda(p_\theta(x_{n-1}|x_n, y) - p_\theta(x_{n-1}|x_n, y'))$$

where $\lambda$ is a scale coefficient. In all of our experiments, we set $\lambda = 8$, which was found to yield the highest FID scores on the validation set. Note that the above extrapolation occurs directly on the logits output by $p_\theta$, in contrast to GLIDE (Nichol et al., 2021), which extrapolates the pixel values.

**Training Objective** For completeness we are adding the training objective of the discrete model. The network is trained to minimize the variational lower bound (VLB):

$$\begin{aligned}
\mathcal{L}_{\text{vlb}} &= \mathcal{L}_0 + \mathcal{L}_1 + \cdots + \mathcal{L}_{N-1} + \mathcal{L}_N, \\
\mathcal{L}_0 &= -\log p_\theta(x_0|x_1, f_{img}(I), \text{knn}_{img}(I, k)), \\
\mathcal{L}_{n-1} &= D_{KL}(q(x_{n-1}|x_n, x_0) \,||\, p_\theta(x_{n-1}|x_n, f_{img}(I), \text{knn}_{img}(I, k))), \\
\mathcal{L}_N &= D_{KL}(q(x_N|x_0) \,||\, p(x_N))
\end{aligned} \tag{8}$$

Where $p(\boldsymbol{x}_N)$ is the prior distribution of timestep $N = 100$, $f_{img}(I)$ is the CLIP image embedding, $\text{knn}_{img}(I, k)$ is the $k$ nearest neighbors in the feature space of the image embedding. The full details can be found in Gu et al. (2021).

## 6.8 CONTINUOUS kNN MODEL

We provide additional implementation details for the continuous diffusion model. Additional training details can be found in Tab. 3.

**Decoder.** We followed (Nichol et al., 2021; Ho et al., 2020; Ramesh et al., 2022) and re-implemented a diffusion *U-net* model. Specifically, we modify the architecture described in (Ramesh et al., 2022) by allowing multiple CLIP embeddings as the condition to the model. Since we do not have a paired text-image dataset, we removed the text transformer, and thus the text embedding. In particular, we use 512 convolution channels, 3 residual blocks, 64 heads channels and attention resolution of 32, 16 and 8. Similarly to our discrete model, we trained two models (1)

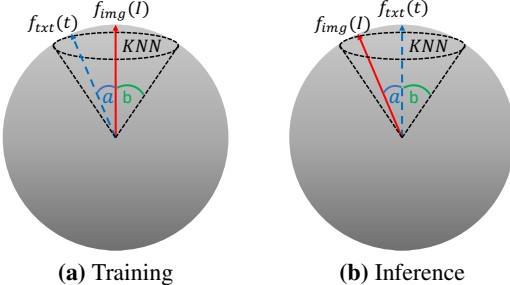

**(a)** Training          **(b)** Inference

**Figure 17:** During training, only the image I is given (red), whereas during inference only the text $t$ is given (blue). In order to bridge the gap between the two distributions during training, we leverage the K nearest neighbors that should have a large enough distribution (dashed cone) to cover the potential text embedding (i.e. $cos(b) < cos(a)$). During inference, the opposite is applied.

a *no-kNN* conditioned only on CLIP image embedding during training, (2) a kNN conditioned on CLIP image embedding and its kNN. Finally, we enable classifier-free guidance by randomly setting the CLIP embeddings to zero 10% of the time. As demonstrated in Tab. 2, we find that humans prefer our model over *no-kNN* 66.8% of the time for image quality and 69.4% of the time for text alignment.

**Super-Resolution.** As the decoder generates images with $64 \times 64$ resolution, we up-sampled the images to $256 \times 256$ using the open-source super resolution of (Nichol et al., 2021). To further up-sample the images to $512 \times 512$ and $1024 \times 1024$ we used the open-source super resolution provided by (Wang et al., 2021).

**Training Objectives** For completeness we are adddding the training objective of our continuous model. Following Ho et al. (2020); Nichol et al. (2021) we are using mean-squared error loss to predict the noise:

$$L := E_{n \sim [1,N], x_0 \sim q(x_0), \epsilon \sim \mathcal{N}(0,\mathbf{I})}[||\epsilon - \epsilon_\theta(x_n, n, y)||^2]$$

where $\epsilon_\theta$ is a $U - net$ model and $y = (f_{img}(x_0), \text{knn}_{img}(x_0, k))$.

|  | Discrete | Continuous |
|---|---|---|
| Number of nearest neighbors | 10 | 10 |
| Diffusion steps | 100 | 1000 |
| Noise schedule | - | cosine |
| Sampling steps | 100 | 250 |
| Model size | 400M | 1B |
| Sampling variance method | - | analytic |
| Dropout | - | 0.1 |
| Weight decay | 4.5e-2 | - |
| Batch size | 512 | 1600 |
| Iterations | 150K | 500K |
| Learning rate | 4.05-4 | 1.4e-4 |
| optimizer | AdamW | AdamW |
| Adam $\beta_2$ | 0.96 | 0.9999 |
| Adam $\epsilon$ | 1.0e-8 | 1.0e-8 |
| EMA decay | 0.99 | 0.9999 |
| warmup | 5000 | 25000 |
| #GPUs | 128 A100 | 200 A100 |

**Table 3:** Training details of our models

**Algorithm 1** Pseudo-code implementation for the construction of the retrieval model, training and sampling using conditioning kNN.

```
───────────────────── Retrieval model construction ─────────────────────
def training(dataset: train image dataset):                              1
    //inverted index of 50k centroids,                                   2
    //with optimized product quantization to 256B                        3
    idx_cfg = "OPQ256_IVF50000_PQ256x8"                                  4
    index = faiss.index_factory(d, idx_cfg, faiss.METRIC_INNER_PRODUCT)  5
    ivf = faiss.extract_index_ivf(index)                                 6
    clustering_index = faiss.index_cpu_to_all_gpus(faiss.IndexFlatIP(d)) 7
    ivf.clustering_index = clustering_index                              8
    train_dataset = []                                                   9
    for image in random.sample(dataset, 1000000):                       10
        train_dataset.append(CLIP_image_embedding(image))               11
    index.train(train_dataset)                                          12
    for image in dataset:                                               13
        index.add(CLIP_image_encoder(image))                            14
    return index                                                        15

──────────────────────────────── Training ────────────────────────────────
def training(I:FAISS index, image, k:Number of NN, t:timestamp [0, T-1]): 1
    image_encoding = CLIP_image_encoder(image)                           2
    kNN = I.search(image_encoding, k)                                    3
    condition = concatenate([image_encoding, kNN])                       4
    image_T = add_noise(image, t)                                        5
    image_0 = diffusion_model(image_T, t, condition)                     6
    loss = criterion(image0, image)                                      7
    return loss                                                          8
                                                                         9

──────────────────────────────── Sampling ────────────────────────────────
def sampling(I: FAISS index, text, k : Number of NN):                    1
    text_encoding = CLIP_text_encoder(text)                              2
    kNN = I.search(text_encoding, k)                                     3
    condition = concatenate([text_encoding, kNN])                        4
    image = sample_noise(T)                                              5
    for t in [T-1, T-2, ..., 0]:                                         6
        image = diffusion_model(image, t, condition)                     7
    return image                                                         8
```

## 6.9 TEXT-ONLY IMAGE MANIPULATION

Our approach is illustrated in Fig. 18. Additional manipulation examples are provided in Figs. 20. The full comparison with the baselines is provided in Fig. 21 and 22. We also provide in Fig. 19 several examples for the process of the manipulated images construction.

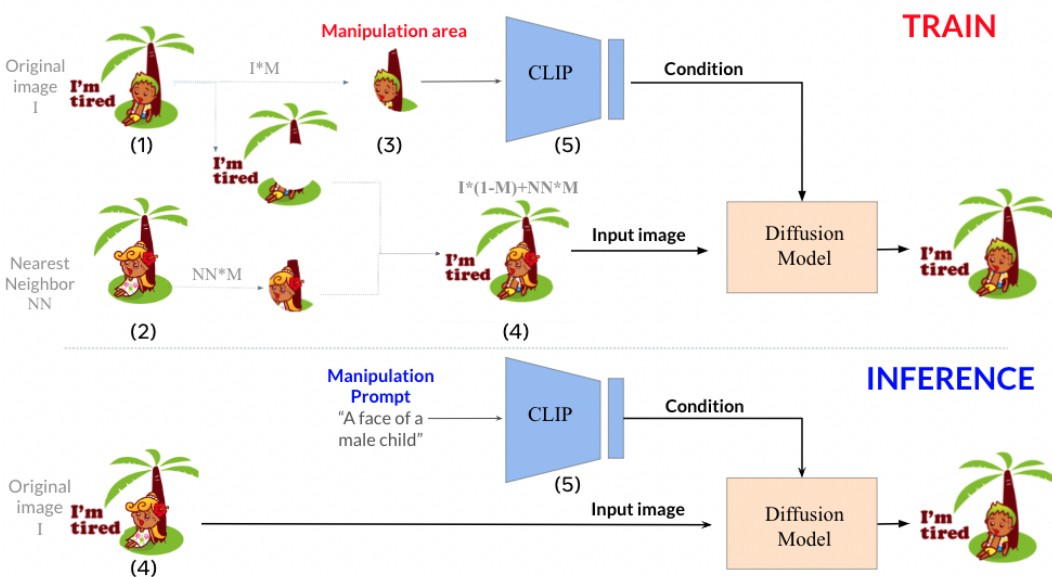

**Figure 18:** An illustration of our manipulation approach. **During training:** Given a training image (1), the model extracts its first nearest neighbor (2). Next, a random local area in the training image is selected (3), and the manipulated image is constructed by replacing the area with the corresponding nearest neighbor (4). The model then receives as input the manipulated image and the clip embedding of the local area that needs to be restored (5). **During inference:** Given an input image and a text query "A face of a male child", the model receives as input the image (4) and the clip embedding of the modifying text (5).

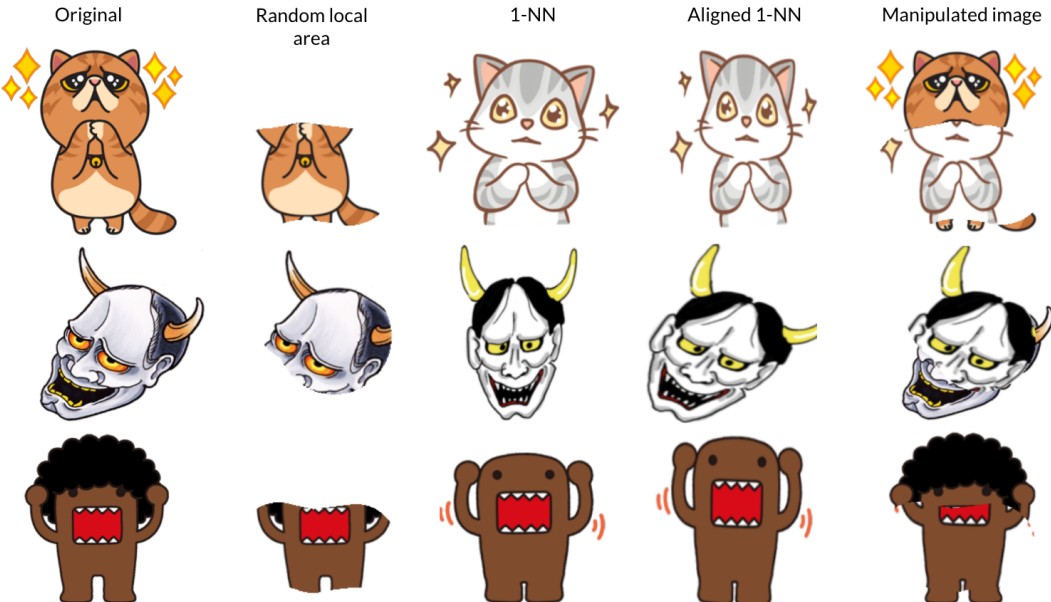

**Figure 19:** Illustration of the manipulated image construction process during training. Given an original image, we select a random local area, and extract the first nearest neighbor (1-NN). Using ECC alignment, we align the nearest neighbor with the original image and replace the random local area with it's corresponding nearest neighbor local area. The model then receives as input the manipulated image, together with the CLIP embedding of the local area, and tries to predict the original image.

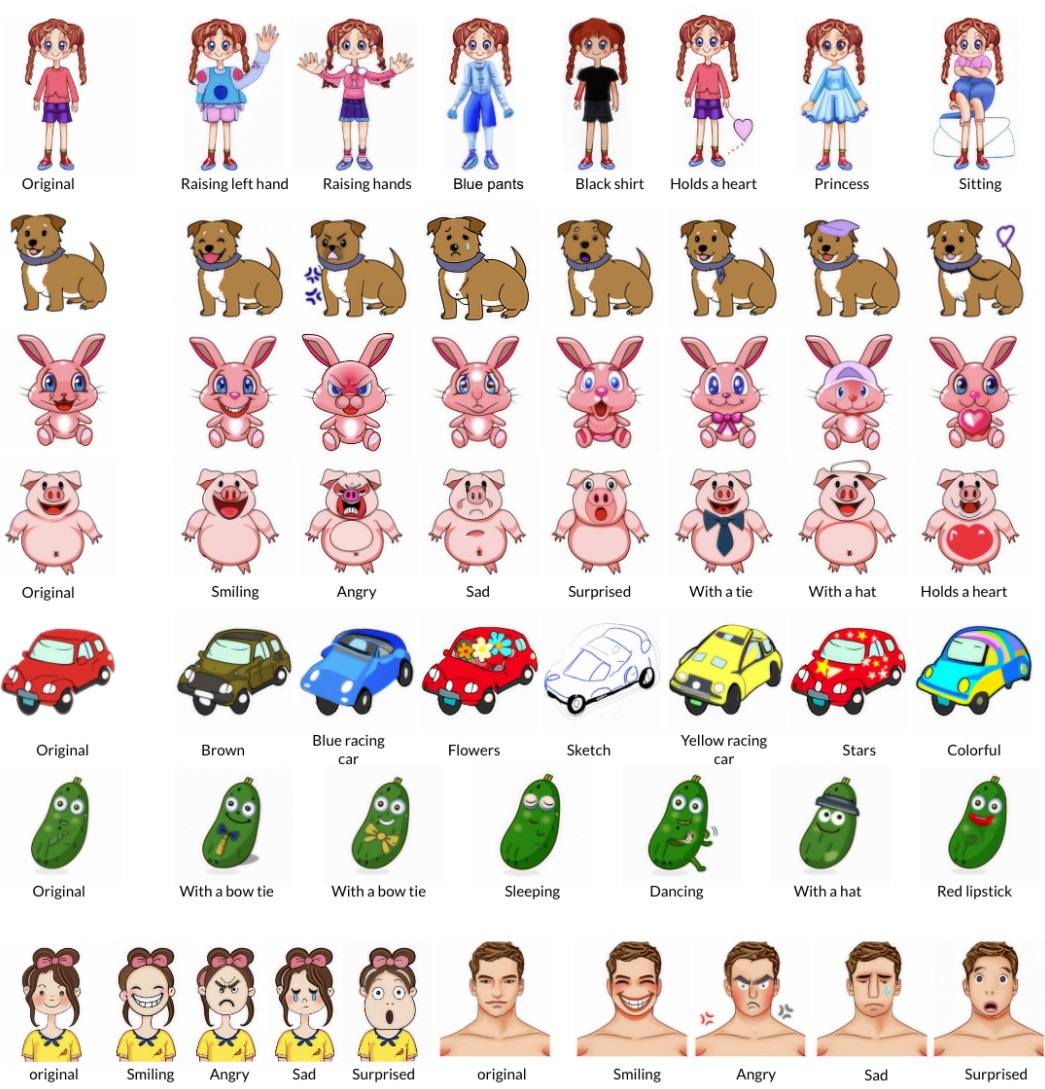

**Figure 20:** Additional manipulation examples, generated using our model.

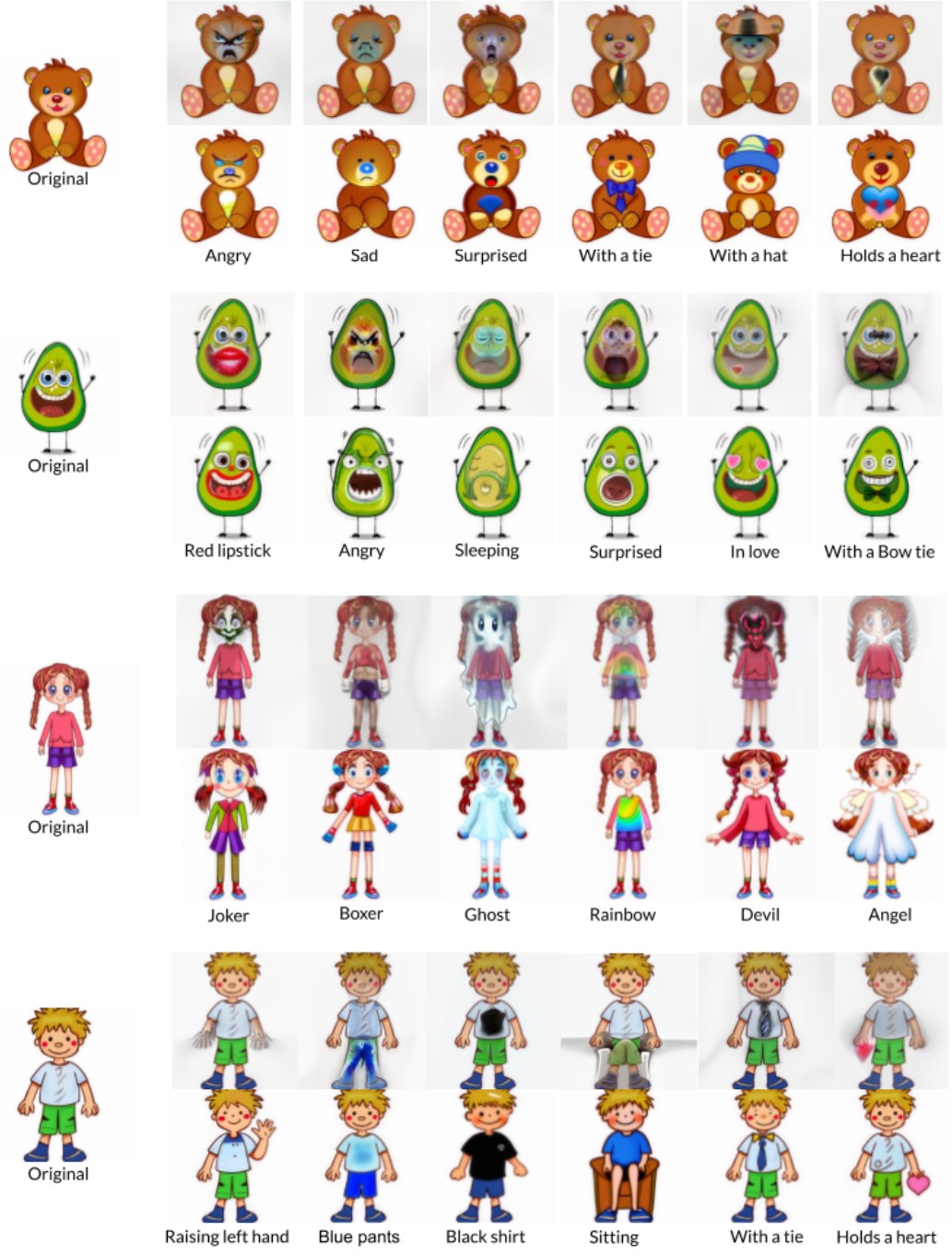

**Figure 21:** comparison to Text2LIVE (Bar-Tal et al., 2022). For each input image, the bottom row corresponds to images generated by our model, and the top row corresponds to images generated by the Text2LIVE model.

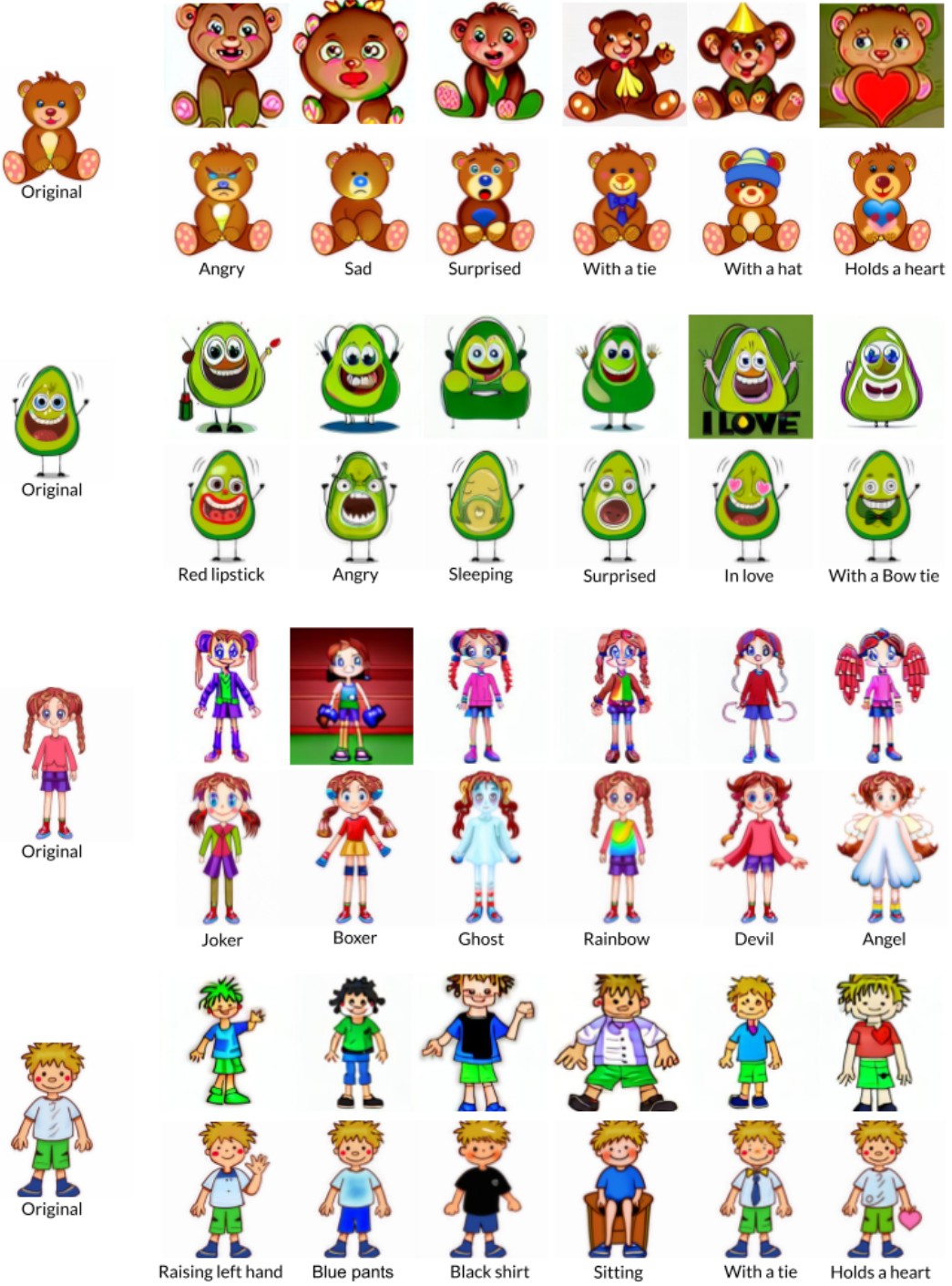

**Figure 22:** comparison to Textual Inversion (Gal et al., 2022). For each input image, the bottom row corresponds to images generated by our model, and the top row corresponds to images generated by the Textual Inversion model.

