# OpenReview forum: "kNN-Diffusion: Image Generation via Large-Scale Retrieval"
_ICLR.cc/2023/Conference — ICLR 2023 poster_

### Official Review · Reviewer_o44L · 2022-10-22

**Confidence:** 3
**Correctness:** 3
**Technical Novelty And Significance:** 2
**Empirical Novelty And Significance:** 2
**Recommendation:** 6

**Clarity, Quality, Novelty And Reproducibility:**

Quality: 8/10

Clarity: 8/10

Novelty: 5/10

Reproducibility: 8/10

**Strength And Weaknesses:**

Pros:

-  The method framework is well motivated to sample from the out-of-distribution images.
-  Competitive results compared with several baselines.
-  Related works are quite thorough for new beginners.

Cons:

- Writing is a little rush, the preliminary knowledge of diffusion model is missing, which makes the method part is very unfriendly to novice of this area.

- Why KNN can solve text-to-image models annotation problem in abstract? It seems that your method is condition on retrieved features instead of text/caption.
- Figure 3 is too rush, what is MMMM, 9143? I am totally missed about what you are trying to express in this figure. The data flow is full of confusion.
- Evaluation details are missing, how to generate condition in your sampling process? Where do those conditioned text/image come from?  Where is the reference set from? How many samples are used to calculate the FID in Table 1 and 2?
- Training and network details are missing, how many parameters in your method, is the parameter number almost the same between your method and the proposed baselines. How many GPUs did you use? Learning rate? optimizer? batch size? epoch number? training time? weight decay?


Minor comments:

- The full comparison is available in the supplement.
- The dataset intro should go to experiments part.
- Figure 6,9 is saved as vector image, pdf.
- The font size of Figure 6 is too small.



**Summary Of The Paper:**

This paper proposes a method to train a conditional diffusion models by removing the necessary of annotation pairs. More detailed, they utilize the kNN retrieved image embedding as condition signal, and can further utilize caption embedding due to the almost-perfect joint distribution between text and images.  The framework also facilitates the generation of out-of-distribution sampling. One extra bonus is they can remove the necessity of mask annotation when conducting local semantic manipulations.

Experimental results demonstrate effectiveness of the proposed method on zero-shot text-to-image generation, the metric is FID and user study around image quality and text-alignment.

**Summary Of The Review:**

The motivation is well consolidated, while too many details about training/inference/preliminary knowledge are missing. Also, the paper is not easy to follow. I suggest the authors to spend more time about the writing of this paper.

=======

post-review update:

The paper has been improved a lot based on my concerns about description, preliminary background, training details, etc. I think the quality of current version is enough for ICLR. I also share similar concern about the marginal novelty of the proposed methods with other reviewers.  Since advantages outweigh disadvantages, I upgraded my result.

---

> ### Author Response · Authors · 2022-11-10
> **Author response (part i out of ii)**
>
> We thank reviewer o44L for the detailed feedback and the useful suggestions, and we hope our response will fully address the concerns raised. We would be happy to further clarify or answer any other questions the reviewer may have.
>
> Please note that there are two parts for the answer.
>
>
> **[W1] preliminary knowledge of diffusion model**
>
> Thank you for your suggestion. An introduction to discrete and continuous diffusion models has been added to the supplementary (Sec. 6.1).
>
> **[W2] Why kNN solve the text-to-image-models annotation problem**
>
> As CLIP provides a joint latent space for text and images, it makes sense to train text-to-image models with CLIP image embedding and infer using CLIP text embedding. However, there is no 1-to-1 mapping between text CLIP embedding and image CLIP embedding. This observation was previously discussed in LAFITE, and is demonstrated in Tab.1-2, Fig.5, Fig.15. The gap between the distributions is explained by the fact that one image can be represented by many texts and vice versa. Therefore, in order to overcome this challenge, we propose extending the image embedding condition to cover possible text embeddings as well. In this way, we ensure that both training and inference conditions are from similar distributions (please refer to paragraphs 1+2+“retrieval model” in the method section, paragraphs 3+4 in the introduction section for more information). Through extensive experiments, we demonstrate that our kNN approach bridges the gap between the distributions:
>
> * For both datasets, photo-realistic and stickers, we trained two models - one conditioned only on CLIP embedding, and the other on both CLIP embedding and kNN embeddings. In both datasets, the FID and the human evaluation score are higher for models trained with kNN. (Tab.1-2).
>
> * A comparison of images generated using our model and the same model without kNN is provided. It can be seen that the images generated by the model without kNN are of lower quality and less aligned with the text (Fig. 5, Fig. 15).
>
> In addition, we have conducted the following experiment to further demonstrate that the retrieval method bridges the gap between the distributions. We have extracted 500 random text-image pairs from the COCO validation set. For each text-image pair we:
>
> 1. Calculated the cosine similarity between the image CLIP embedding and the text CLIP embedding:
> **dist_0** = cosine_similarity (image_embedding, text_embedding)
> 2. Calculated the cosine similarity between the image CLIP embedding and the average of the text CLIP embedding and its kNN:
> **dist_k** = cosine_similarity(image_embedding, mean(text_embedding, 1NN_embedding, … kNN_embedding).
>
> Then, for both experiments, we calculated the average cosine similarity across all 500 pairs.
> The average score in (1) represents the similarity between the images and texts distribution: **avg_dist_0** = $(\sum_1^{500}dist_0)/500$). The average score in (2) represents the similarity between the images and the averaged embedding of the text with its kNN: **avg_dist_k** = $(\sum_1^{500}dist_k)/500$).
>
> We provide the results in the following table:
> |    | text embedding, image embedding (avg_dist_0)  | text embedding+1NN, image embedding  (avg_dist_1)   | text embedding+5NN, image embedding (avg_dist_5)| text embedding+10NN, image embedding (avg_dist_10) | text embedding+20NN, image embedding (avg_dist_20) |
> | ---------------------------   | ------------------        |  -----------      | ---------      | ------------ |    ------------ |
> | cosine similarity score   | 0.3 +- 0.03 | 0.5 +- 0.06 | 0.64 +- 0.08 | 0.67 +- 0.08 | 0.69 +- 0.09 |
>
> **It can be seen that the gap between the two distributions decreases as the number of kNN increases.**
>
> We also included in the revised paper a tSNE visualization of the CLIP image and text distributions (of 500 pairs) when using a different number of nearest neighbors (see Fig. 8). As can be seen from the leftmost subfigure, there is a gap between the distributions when no kNN is used. By adding kNN to the mean CLIP embedding of the text, the gap between the distributions decreases, demonstrating the importance of kNN in bridging the distributions.
> ***
>
> **[w3]Figure 3**
>
> We have changed the figure. Please let us know if you still find the new figure confusing.

---

> > ### Author Response · Authors · 2022-11-10
> > **Author response (part ii out of ii)**
> >
> > **[W4] Evaluation details**
> > >  how to generate condition in sampling process
> >
> > During the sampling process, the condition is the clip embedding of the prompt, as well as its kNN retrieved from the retrieval database (please see the second paragraph in the introduction and the "retrieval model" in the method). As shown in Fig. 3, at inference time, the CLIP text embedding is used to retrieve the kNN embeddings. Following the reviewer’s questions, we have revised the supplementary to include pseudocode for the retrieval method construction, the training step and the sampling process (algorithm 1).
> >
> > > conditioned text/image, reference set from, number of samples for FID
> >
> > As mentioned in Sec. 4.1, we followed LAFITE evaluation protocol when evaluating our model on MS-COCO, CUB and LN-COCO. In particular, we used 30,000 images to calculate the FID for MS-COCO. We have added this number to the paragraph. The human evaluation was conducted on 600 images pairs, as mentioned in the "dataset and metrics" paragraph. To calculate the FID on the stickers dataset, we used 3,000 images generated from ClipCap auto-generated captions, as described in the last paragraph of the text-to-sticker generation section.\
> > For each sample, the condition is the text clip embedding and the kNN embeddings retrieved from the training retrieval database.
> > ***
> > **[W5] Training and network details**
> >
> > Thank you. In both methods we followed the baselines in their training parameters, in order to emphasize the contribution of the kNN conditioning to the results (see the last sentence in Sec. 6.6 and the first sentence in the “Decoder” paragraph in Sec. 6.7). Following the reviewer’s questions, we have revised the supplementary material to include the implementation details of the discrete and continuous diffusion models (Sec. 6.6-6.7) and table with all the details and parameter settings (Tab.3).
> > If you find anything missing, please let us know. We would be happy to provide any additional clarifications you may require.
> >
> > > is the parameter number almost the same between your method and the proposed baselines.
> >
> > As can be seen in Fig.6 in the paper, our photo-realistic model contains 400M parameters, and it is substantially smaller than the supervised baselines, although having comparable results.
> >
> > **Minor comments**
> >
> > Thank you for your suggestions. We have updated the revised manuscript in accordance with your comments.

---

> ### Author Response · Authors · 2022-11-20
> **Rebuttal period ending–we appreciate your feedback!**
>
> Dear reviewer o44L,
>
> As the rebuttal/author discussion period is closing, we sincerely look forward to your feedback. We believe that our responses fully address all raised concerns, and would highly appreciate your feedback.
>
> [comments]
>
> 1) We kindly refer the reviewer to "General response (part 2)" for our contributions summary and the main revisions summary.
> 2) We are in the process of releasing the code before the camera-ready. Meanwhile, we added the pseudocode for (1) constructing the retrieval method (2) training with kNN conditioning, (3) sampling with kNN conditioning, and added additional training and implementation details to the supplementary material (Tab. 3, Sec. 6.6-6.7).
>
> Please let us know if there are any further questions or comments about this paper. We strive to consistently improve the paper and it would be our pleasure to have your precious feedback!
>
> Kind Regards,\
> The authors

---

> ### Author Response · Authors · 2022-12-02
> **A kind reminder to reviewer o44L**
>
> Dear reviewer o44L,
>
> We would like to thank you again for your time in reviewing our work. As the deadline for discussion is approaching, we really hope to have a further discussion with you. We believe we have addressed your concerns. In particular:
> * Further demonstrated "Why KNN can solve annotation problem”
> * Further demonstrated the kNN contribution by adding scalability analysis of our model. (Sec. 4.3)
> * Revisited Fig.3 of our architecture to reflect reviewers' comments
> * Added preliminary knowledge on diffusion models, pseudocode for (1) training, (2) sampling and (3) retrieval database construction. In addition, provided additional training and implementation details.
>
> We therefore believe that we have made significant contributions to several practical problems: (1) training small and efficient text-to-image models without an explicit paired text-image dataset, (2) performing local and semantic image manipulations without masks, and without optimization; we present a novel approach that outperforms the recent baselines, and is much faster **(only 8 seconds for each manipulation compared to two hours for Textual Inversion and 9 minutes for Text2LIVE)**, (3) zero-shot out of distribution generation.
>
> We would  be happy to hear your feedback and provide more clarification if necessary.
>
> Thank you,
>
> The authors

---

### Official Review · Reviewer_myk8 · 2022-10-24

**Confidence:** 2
**Correctness:** 3
**Technical Novelty And Significance:** 3
**Empirical Novelty And Significance:** 3
**Recommendation:** 5

**Clarity, Quality, Novelty And Reproducibility:**

Given the limited motivation and technical novelty of the work, I don’t think the current version meets the standard of the ICLR.

**Strength And Weaknesses:**

Strengths:
1.	The paper is generally well-written and easy to follow.
2.	The paper introduces an easy approach to facilitate the training of text-free text-to-image generation.
3.	The experiments are comprehensive, it demonstrates the strength of the proposed method from different perspectives.
Weakness:
1.	The technical novelty might be limited, its model is conceptually similar to several previous works (e.g., CLIP, CogView, and LAFITE). The main contribution lies in the training strategy (kNN Retrieval) rather than in the techniques.
2.	The Retrieval Model (Section 3) is described very vaguely. The provided details are not sufficient to understand the module and re-implement it.
3.	It would be also helpful to release the code and pre-trained model for subsequent research.

**Summary Of The Paper:**

This paper presents an approach to improve the training performance and overall quality of the recently introduced text-to-image generative framework. By leveraging a kNN-based retrieval method, the proposed method 1) adds more flexibility to the training data, 2) greatly reduces the computational cost. Extensive experiments and ablation studies are conducted to prove the efficacy and the efficiency of the model.
The description of the technical approach is confusing and vague. In Section 3, the authors firstly criticize the use of CLIP embedding since it alone cannot accurately bridge the gap between the text and image distributions. However, the authors still choose to use the CLIP pretrained model to do the job without any modifications. Without any proper justifications on how the proposed retrieval model could solve the problem, the practical motivation of the paper is lacking.

**Summary Of The Review:**

This is a good paper with promising experimental results, with some aspects of the presentation that should be improved and clarified. Also, I have the following questions regarding to the paper.

Questions to Authors:
1.	Could the authors provide more justifications on how could adding the retrieval part bridge the gap between the text and image distributions?
2.	How does the encoders map text descriptions and image samples to a joint feature space? Do you merely use the pretrained models?
3.	The subsection of “Image generation network” also lacks sufficient details. How does the generation network generate new images with conditioned on other similar instances? The equations in the subsection don’t indicate any signals that this is a conditional generation framework.
4.	How does the kNN search avoid the problem of “curse of dimensionality”?
5.	Without any details of the parameter setting and providing model details, the experimental merit is also hard to evaluate.
6.	The scalability of the model is highlighted as one of the primary contributions. However, the corresponding scalability analysis experiment is not found in the paper.

---

> ### Author Response · Authors · 2022-11-10
> **Authors response (part i out of iii)**
>
> We would like to thank reviewer myk8 for the detailed feedback and useful suggestions. The following are our responses to the reviewer's questions. We hope that we have addressed your concerns. If not, please let us know and we will be happy to answer any further questions you may have.
>
> Please note that there are three parts for the answer.
>
> ***
> **[Q1] more justifications on how could adding the retrieval part bridge the gap between the text and image distributions**
>
> As CLIP provides a joint latent space for text and images, it makes sense to train text-to-image models with CLIP image embedding and infer using CLIP text embedding. However, there is no 1-to-1 mapping between text CLIP embedding and image CLIP embedding. The gap between the distributions was previously discussed in LAFITE and can explained by the fact that one image can be represented by many texts and vice versa. In order to overcome this challenge, we propose extending the image embedding condition to cover possible text embeddings as well. In this way, we ensure that both training and inference conditions are from similar distributions. Through extensive experiments, we demonstrate that our kNN approach bridges the gap between the distributions:
>
> * For both datasets, photo-realistic and stickers, we trained two models - one conditioned only on CLIP embedding, and the other on both CLIP embedding and kNN embeddings. In both datasets, the FID and the human evaluation score are higher for models trained with kNN. (Tab.1-2).
>
> * A comparison of images generated using our model and the same model without kNN is provided. It can be seen that the images generated by the model without kNN are of lower quality and less aligned with the text (Fig. 5, Fig. 15).
>
> Based on the reviewer's comments, we have conducted the following experiment to further demonstrate that the retrieval method bridges the gap between the distributions. We have extracted 500 random text-image pairs from the COCO validation set. For each text-image pair we:
>
> 1. Calculated the cosine similarity between the image CLIP embedding and the text CLIP embedding:
> **dist_0** = cosine_similarity (image_embedding, text_embedding)
> 2. Calculated the cosine similarity between the image CLIP embedding and the average of the text CLIP embedding and its kNN:
> **dist_k** = cosine_similarity(image_embedding, mean(text_embedding, 1NN_embedding, … kNN_embedding).
>
> Then, for both experiments, we calculated the average cosine similarity across all 500 pairs.
> The average score in (1) represents the similarity between the images and texts distribution: **avg_dist_0** = $(\sum_1^{500}dist_0)/500$). The average score in (2) represents the similarity between the images and the averaged embedding of the text with its kNN: **avg_dist_k** = $(\sum_1^{500}dist_k)/500$).
>
> We provide the results in the following table:
> |    | text embedding, image embedding (avg_dist_0)  | text embedding+1NN, image embedding  (avg_dist_1)   | text embedding+5NN, image embedding (avg_dist_5)| text embedding+10NN, image embedding (avg_dist_10) | text embedding+20NN, image embedding (avg_dist_20) |
> | ---------------------------   | ------------------        |  -----------      | ---------      | ------------ |    ------------ |
> | cosine similarity score   | 0.3 +- 0.03 | 0.5 +- 0.06 | 0.64 +- 0.08 | 0.67 +- 0.08 | 0.69 +- 0.09 |
>
> **It can be seen that the gap between the two distributions decreases as the number of kNN increases.**
>
> As an additional demonstration of this observation, we have included in the revised paper a tSNE visualization of the CLIP image and text distributions (of 500 pairs) when using a different number of nearest neighbors (see Fig. 8). As can be seen from the leftmost subfigure, there is a gap between the distributions when no kNN is used. By adding kNN to the mean CLIP embedding of the text, the gap between the distributions decreases, demonstrating the importance of kNN in bridging the distributions.
> ***
> **[Q2] encoders map text and image to a joint feature space**
>
> As described in the “retrieval model” section, we use a pre-trained CLIP text encoder as our text encoder, and a pre-trained CLIP image encoder as our image encoder. CLIP is a multi-modal text-image encoder that maps text-image pairs into the same latent space. It has been trained with a contrastive loss, aiming to minimize the cosine distance between matching text-image pairs, while maximizing the cosine distance between non-matching text-image pairs. In both training and inference, our model is conditioned on CLIP embeddings.
> We have revised Fig.3 to further clarify the model architecture.

---

> > ### Author Response · Authors · 2022-11-10
> > **Author response (part ii out of iii)**
> >
> > **[Q3] “Image generation network”**
> >
> > As described in the method (first, second, and the “retrieval model” paragraphs), during training, we condition the model on (1) image CLIP embedding (2) its kNN image embeddings. At inference time, given an input text, we condition the model on (1) text CLIP embedding, (2) its kNN image embeddings.
> > To further clarify the generation process, we added to the supplementary the pseudocode for (1) training with kNN conditioning and (2) sampling with kNN conditioning (algorithm 1).
> >
> > In addition, the equations of the conditional network are provided in Sec. 6.6 (“training objective”) and in Sec. 6.7 (“training objective”). The condition is also provided in the discrete equation in the “image generation network” paragraph.  In all these equations, “y” represents the condition. Following the review, we also added the condition to the continuous equation. In addition, we revised Sec. 6.6-7 to further clarify the use of the condition.
> > ***
> > **[Q4] “curse of dimensionality” problem**
> >
> > We thank the reviewer for highlighting this point. The "curse of dimensionality" was not observed in our experiments (see Fig. 1-2, Fig. 5, Fig. 15-19).
> > “Curse of dimensionality problem” refers to the two difficulties mentioned [here](https://www.baeldung.com/cs/k-nearest-neighbors#high-dimensionality), and we address both difficulties below:
> >
> > > “the assumption of similar points being situated closely breaks”
> >
> > We did not face this difficulty since the feature space in which we are applying kNN search has been trained by CLIP to match semantically similar image-text pairs based on cosine similarity (which is also the similarity we are using to find the kNN). CLIP was trained on a large dataset of 400M samples in order to maximize the cosine similarity of semantically closed text-image pairs and minimize the cosine similarity of other pairs. As a result, the data distribution in this feature space is not uniform, and the distances are meaningful.
> >
> > > “computationally more expensive to compute distance and find the nearest neighbors”
> >
> > As described in Sec. 6.5, we are using FAISS to perform efficient kNN search. FAISS is an effective similarity search tool that supports billions of high-dimensional data points. Through the use of clustering and quantization techniques, FAISS addresses the "curse of dimensionality" problem. In particular: (i) we are using non-exhaustive search, (ii) we compredss our vectors using Product quantization (PQ) technique (please refer to Sec. 6.5 for more details). As a result, it takes 0.0067 += 0.002 seconds to perform a kNN search for one query on our 400M large scale index.
> > ***
> > **[Q5] parameter setting, model details.**
> >
> > Following the reviewer’s questions, we have revised the supplementary material to include additional training details and implementation details. In particular:
> > * Additional implementation details for the discrete and continuous models (Sec. 6.6-6.7).
> > * A table listing all the details of the training (Tab.3).
> > * Pseudocode for (1) training, (2) sampling, and (3) retrieval database construction (algorithm 1).
> > ***
> > **[Q6] scalability analysis**
> >
> >  ***== See also the updated answer in the last comment ==***
> >
> > One of the primary contributions of our kNN method is that it enables comparable generation results to much larger models trained on text-image datasets by using a significantly smaller model trained on images-only dataset (as described in the third paragraph of the intro for example). We would appreciate it if you could let us know if "scalability" refers to this contribution.
> >
> > This contribution was demonstrated as follows:
> > Fig. 6 shows that while our 400M parameters model was trained on an image-only dataset, it achieves comparable results to larger recent models trained with full text-image pairs (e.g. it outperforms latent diffusion which uses 1.5B parameters). In addition, it outperforms models trained with the same number of parameters, but without kNN and with full text-image pairs, such as the recent Parti-350M and also a variant of our method without kNN. Finally, it outperforms recent models trained on an image-only dataset. This observation is illustrated in Fig. 6, where our model resides in the area of small FID as the supervised baselines, as well as in the area of the small number of parameters as the unsupervised models, therefore benefiting from both worlds.
> >
> > Following the reviewer's question, and to further demonstrate this observation, we conducted an experiment in which we trained a small model of 250M parameters with kNN and compared it to a 400M parameter model trained without kNN (“no-kNN”). Both models were trained without any text. The smaller model with kNN achieves a FID score of 17.6, whereas the larger model without kNN achieves a score of 32.8. Therefore, It can be seen that kNN enables us to use smaller models while maintaining the same or even better performance.

---

> > > ### Author Response · Authors · 2022-11-10
> > > **Author response (part iii out of iii)**
> > >
> > > **[W1] Contribution**
> > >
> > > Our novel approach tackles the following important and practical problems: (1) training small models without an explicit text-image dataset (2) performing local and semantic manipulations without using masks and without requiring optimization process for each input, (3)  generating out-of-distribution images without fine-tuning and without optimization.
> > >
> > > All these are addressed by our generative model. In particular:
> > >
> > > (1) The current SOTA text-to-image models are explicitly trained on large-scale text-image paired datasets. The requirement of paired dataset prevents using the models on many domains where only images are available (e.g. stickers dataset). While both LAFITE and FuseDream tried to solve this problem, LAFITE needed to train an additional model on text-image dataset in order to achieve better results, and FuseDream required an optimization process for each input.
> > >
> > > Our work tackles these issues and proposes a simple, but novel approach that can be applied to different backbones, and for each one of them improve their results (Tab.2). Using extensive experiments (Tab.1-2, Fig.2, Fig.5, Fig.15-19) we show that leveraging kNN enables training models without explicit text-image pairs, while producing high quality results even with a significantly smaller number of parameters.
> > >
> > > (2) Current manipulation techniques are either (a) limited to global editing; (b) rely on masks; (c) don’t preserve identity; or, (d) require optimization for each input.
> > >
> > > We propose a novel approach to perform semantic and local manipulations, without using masks, and without any optimization. Our generation process time is **only 8 seconds**. This is a significant improvement compared to the recent Textual inversion which takes 2 hours and Text2Live which takes 9 minutes . In addition, compared to the baselines, our model is capable of performing challenging manipulations, while preserving the identity of the object (Fig.1, Fig. 4, Fig.20-22).
> > >
> > > (3) We demonstrate out-of-distribution generation capabilities without fine-tuning (Fig.7, Fig.9). Using a model trained with a retrieval method on a specific index, we demonstrate how one can generate images of different distributions and styles that the model was not trained on, by changing only the index, without fine-tuning.
> > > ***
> > > **[W2] Re-implementation of the Retrieval Model**
> > >
> > > Following the reviewer’s request, we have revised the retrieval method section in the supplementary to further describe our use of the FAISS method (Sec. 6.5). To enable re-implementation of the model, we added pseudocode describing the process of retrieval method construction (algorithm 1).
> > > ***
> > > **[W3] open-source code**
> > >
> > > We are in the process of releasing the code for our paper before the camera-ready. Meanwhile, we added the pseudocode for (1) constructing the retrieval method (2) training with kNN conditioning, (3) sampling with kNN conditioning (see algorithm 1 in the supplementary). Further details regarding training and implementation have been added to the supplementary material (Tab. 3, Sec. 6.6-6.7).
> > > ***
> > > >  "In Section 3, the authors firstly criticize the use of CLIP embedding since it alone cannot accurately bridge the gap between the text and image distributions. However, the authors still choose to use the CLIP pretrained model to do the job without any modifications."
> > >
> > > Throughout the paper we discuss the challenge of using only CLIP embeddings to bridge the gap between the distributions. In the introduction section we (1) present previous works that tried to use CLIP to train models without text (first paragraph), (2) present our remedy for this challenge by extending the condition distribution (third paragraph). We then further emphasize the problem in the related work section (We kindly refer the user to the last row of "text-to-image models" in related work).
> > >
> > > As we explain in detail in the answer of your first question above ([Q1]), the gap between the distributions is not due to CLIP being a “weak encoder”. As a matter of fact we agree with [1] saying that  *“CLIP  has emerged as a successful representation learner for images...they are robust to image distribution shift, have impressive zero-shot capabilities, and have been fine-tuned to achieve state-of-the-art results on a wide variety of vision and language tasks”*. We hypothesize that the problem arises because there is no 1-to-1 mapping between each image and text.
> > > Our modification tackles this specific issue by extending the condition distribution to cover the potential text embedding at test time. In particular, Tab. 1-2 demonstrate the gain in performance achieved when leveraging the kNN model. This gain is further demonstrated using qualitative comparisons (Fig.2, Fig.15, Fig.17). Lastly, Fig. 8 visualizes the gain of using kNN in bridging the distribution gap.
> > >
> > > *[1] Ramesh, Aditya, et al. "Hierarchical text-conditional image generation with clip latents." arXiv preprint arXiv:2204.06125 (2022).*

---

> > > > ### Author Response · Authors · 2022-11-14
> > > > **Scalability analysis graph**
> > > >
> > > > We have just posted a new revision containing the full results of the scalability analysis experiment.
> > > > Following the reviewer's comment, we trained three additional models for both settings - with, and without kNN. The first one with 250M parameters, the second with 150M parameters and the third with 35M parameters.
> > > >
> > > > We have reported the results in Fig.11 (of the revised paper).
> > > >
> > > > As can be seen, adding kNN to the model consistently improves performance for all model sizes.
> > > > Furthermore, a performance improvement can be achieved using much smaller models with kNN. For example, the 35M parameters model trained with kNN outperforms the 400M model trained without kNN.
> > > >
> > > > We will be happy to answer any further questions the reviewer may have.

---

> ### Author Response · Authors · 2022-11-20
> **Follow-up with reviewer myk8**
>
> Dear reviewer myk8,
>
> Thank you again for the detailed feedback and useful suggestions which greatly improved our paper.
>
> We would respectfully like to follow up to see if our response addresses your concerns. If there is a question or concern that remains, please let us know so we can properly address it.
>
> Thank you,\
> The authors

---

> ### Author Response · Authors · 2022-12-02
> **A kind reminder to reviewer myk8**
>
> Dear reviewer myk8,
>
> We would like to thank you again for your time in reviewing our work. As the deadline for discussion is approaching, we really hope to have a further discussion with you. We believe we have addressed your concerns.
>
> We believe that we have made significant contributions to several practical problems:
> 1)  training small and efficient text-to-image models without an explicit paired text-image dataset,
> 2)  performing local and semantic image manipulations without masks, and without optimization; **we present a novel approach that outperforms the recent baselines, and is much faster - only 8 seconds for each manipulation compared to two hours for Textual Inversion and 9 minutes for Text2LIVE)**,
> 3)  zero-shot out of distribution generation
>
> We kindly refer the reviewer to the "general response (part 2)" for the detailed summary of our contributions.
>
> We would  be happy to hear your feedback and provide more clarification if necessary.
>
> Thank you,
>
> The authors

---

### Official Review · Reviewer_5XrX · 2022-10-24

**Confidence:** 4
**Correctness:** 3
**Technical Novelty And Significance:** 3
**Empirical Novelty And Significance:** 3
**Recommendation:** 8

**Clarity, Quality, Novelty And Reproducibility:**

From novelty point of view, it seems that most of the building blocks of the solution were known before (the retrieval part, the generation part). Maybe the image manipulation method stands out; at least I never saw a similar idea.

I also would recommend discussing paper on semi-parametric image generation which used raw pixels as input, like Qi, Xiaojuan, et al. "Semi-parametric image synthesis." CVPR2018 or Iskakov, Karim. "Semi-parametric image inpainting." arXiv preprint arXiv:1807.02855 (2018).

**Strength And Weaknesses:**

Strengths:
1) The paper presentation is overall good. Illustrations make sense, most of the ideas are clearly expressed.
2) The results look good, both FID and user studies. I was a bit surprised that having just embeddings as input (without the access to raw images) is enough to make a big difference. I guess the neighbors provide the hint of how far the query is from the real images, so it allows the model to prefer directions to the real embeddings.

Weaknesses:
1) A big red flag for me is the claim that no text pairs were used during the training. To reproduce the model from data one would have to pretrain the encoders on the paired data, so this claim is misleading. I highly recommend to put it into correct context to avoid reader confusion.
2) There are some details missing. E.g. I could not find the value of K during training, which should help interpret Fig. 9 (maybe the optimal number of neighbors is just the train one?)
3) Not clear whether the solution is going to be open-sourced.


Misc: Sometimes there is kNN in the text/figures, sometimes it is KNN, sometimes knn. Sometimes it is k, sometimes K, sometimes it is kNN. It also makes sense to explicitly state that Image quality and Text-alignment are the human evaluation metrics (e.g. it is mentioned for table 2, but not for table 1).

**Summary Of The Paper:**

The paper proposes a method for generating images from text with an addition of a retrieval component. Authors use pretrained text and image encoders and train diffusion-based generative models that use an image embedding and a set of image embeddings closest to the target one to reconstruct the image. During inference, instead of the image embedding, a text embedding is used (which comes from the frozen clip encoder, pretrained on image-text pairs). Authors demonstrate that the retrieval component is crucial for image quality both in online and in human studies and also extend the functionality for global image manipulation, which looks rather convincing.

**Summary Of The Review:**

Overall, I think this is a decent paper. I my main criticism is about claim "training ... model without any text" which might very misleading. I believe it has to be fixed.

Update after the discussion - my comments were address during the rebuttal period, so I updating the rating (given the code will be open-sourced).
Also I suggest the authors to update Figure 3 - the kNN part there is hard to understand (both what it does and what is the output)

---

> ### Author Response · Authors · 2022-11-10
> **Author response**
>
> We thank reviewer 5XrX for the comprehensive review, important comments and points for discussion. The attention to details and the positive feedback regarding our novel manipulation approach are greatly appreciated. Please find below the answers to your questions. Upon request, we will be happy to provide further clarifications in order to address your concerns.
>  ***
> **[W1] no text pairs were used during the training**
>
> Thank you for your comment. We agree that this is a subtle point that needs to be highlighted and clarified. We followed previous work [1] to define the task of "training text-to-image models without text". Similar to us, previous work that did not explicitly use paired text-image dataset, utilized a pre-trained CLIP as a shared representation space: FuseDream performed a direct optimization to a pre-trained model using CLIP as a component in its loss, and LAFITE proposed training with CLIP’s image embedding while inferring with CLIP’s text embedding. All of these works are considered in the setting of models trained without text data.
>
> In order to add clarity, we have edited the introduction section in the revised version to include your comment (in particular: “training a text-to-image model using only pre-trained multimodal embeddings but without an explicit text-image dataset”).
> Upon request, we will be happy to provide additional clarifications.
>
> *[1] Zhou, Yufan, et al. "Lafite: Towards language-free training for text-to-image generation." arXiv preprint arXiv:2111.13792 (2021)*
> ***
> **[W2] the value of K during training**
>
> Thank you for your important comment. The number of nearest neighbors was chosen to be k=10, and this value was added to the method section of the revised version (see the end of the retrieval model component).
> ***
> **[W3] open-source code**
>
> We plan to release the code of our paper before the camera-ready. Meanwhile, we have added to the supplementary the pseudocode for (1) constructing the retrieval method (2) training with kNN conditioning, and (3) sampling new generations (see algorithm 1). In addition, we added a table with the training details to the supplementary (Tab. 3).
> ***
> **Novelty**
>
> Our novel approach tackles the following important and practical problems: (1) training small models without an explicit text-image dataset (2) performing local and semantic manipulations without using masks and without requiring optimization process for each input, (3)  generating out-of-distribution images without fine-tuning and without optimization.
>
> All these are addressed by our generative model. In particular:
>
> (1) The current SOTA text-to-image models are explicitly trained on large-scale text-image paired datasets. The requirement of paired dataset prevents using the models on many domains where only images are available (e.g. stickers dataset). While both LAFITE and FuseDream tried to solve this problem, LAFITE needed to train an additional model on text-image dataset in order to achieve better results, and FuseDream required an optimization process for each input.
>
> Our work tackles these issues and proposes a simple, but novel approach that can be applied to different backbones, and for each one of them improve their results (Tab.2). Using extensive experiments (Tab.1-2, Fig.2, Fig.5, Fig.15, Fig.17) we show that leveraging kNN enables training models without explicit text-image pairs, while producing high quality results even with a significantly smaller number of parameters.
>
> (2) Current manipulation techniques are either (a) limited to global editing; (b) rely on masks; (c) don’t preserve identity; or, (d) require optimization for each input.
>
> We propose a novel approach to perform semantic and local manipulations, without using masks, and without any optimization. Our generation process time is **only 8 seconds**. This is a significant improvement compared to the recent Textual inversion which takes 2 hours and Text2Live which takes 9 minutes . In addition, compared to the baselines, our model is capable of performing challenging manipulations, while preserving the identity of the object (Fig. 4, Fig.20-22).
>
> (3) We demonstrate out-of-distribution generation capabilities without fine-tuning (Fig.7, Fig.9). Using a model trained with a retrieval method on a specific index, we demonstrate how one can generate images of different distributions and styles that the model was not trained on, by changing only the index, without fine-tuning.
> ***
> **semi parametric models discussion**
>
> We appreciate you bringing these relevant papers to our attention. Our related work has been updated to include them.
> ***
> **Misc**
>
> Thank you for your feedback. The revision has been updated accordingly.

---

> > ### Comment · Reviewer_5XrX · 2022-11-10
> > **Abstract still has misleading claims**
> >
> > Abstract still says "training a substantially small and efficient text-to-image diffusion model without any text" which is not true!
> > It does not matter if other papers did such claims (shame on them). They create an impression there is a way to learn image-to-text correspondence (which is a byproduct of text-to-image generation) without using any text.

---

> > > ### Author Response · Authors · 2022-11-11
> > > **We thank the reviewer for highlighting this.**
> > >
> > > We went through the paper and clarified this. We've updated the abstract as well. Please let us know if any other clarification is needed.

---

### Official Review · Reviewer_yR24 · 2022-10-24

**Confidence:** 3
**Correctness:** 3
**Technical Novelty And Significance:** 2
**Empirical Novelty And Significance:** 3
**Recommendation:** 6

**Clarity, Quality, Novelty And Reproducibility:**

The paper is easy to follow.
Quality is okay, it could use more experiments.
The method is not novel but the application is pretty interesting.

**Strength And Weaknesses:**

Strengths
- [S1] The application of retrieval is interesting and tackling limited training data is a practical problem.
- [S2] The controlled settings show strong performance.

Weaknesses
- [W1] Benchmark seems weak. Not a lot of baselines, and main results are only zero-shot but it's not clear that this is the focus of the paper.
- [W2] Some drawbacks of retrieval-based methods are not well discussed. For example, the choice of k is highly dataset specific, how would someone select this in practice? The proposed method also do not scale well to larger dataset if every query involves a retrieval against the whole dataset.


**Summary Of The Paper:**

This paper proposes to apply retrieval in text-to-image generation based on an encoder-decoder architecture. K-nearest neighbors of some image database is computed on top of the encoder output, and a transformer-based diffusion network is applied on these neighbors to produce the final output image.

The main idea of the paper is that by leveraging the retrieval process, smaller training dataset can be used, and even out-of-distribution image generation is possible by choosing appropriate database from which to draw the k-nearest neighbors.

In settings controlled for backbones author show significant improvements on common image generation metrics.

**Summary Of The Review:**

I recommend weak accept. The experimentation could be stronger, but the idea and the application is quite interesting. The problem this paper tackles is relevant.

---

> ### Author Response · Authors · 2022-11-10
> **We thank reviewer yR24 for the positive review and for providing important comments.**
>
>  We will be happy to answer any further questions the reviewer may have.
>
> **[W1] Benchmark, baselines, results.**
>
> Thank you for your comment. We followed previous work standard setting ([1,2,3,4,5]) to benchmark our model on MS-COCO validation set, where we compared our model with the current state-of-the-art supervised models trained with paired text-image dataset, such as DALLE2, Imagen, Parti, and Make-a-Scene (Fig.6). Additionally, we compared our model to unsupervised models trained without text (similar to us), such as LAFITE, Fusedream, and a variant of our model without kNN (Tab.1). To the best of our knowledge, LAFITE is the only prior work that has been trained on image-only dataset and provided results on MS-COCO with fine-tuning. In this case, our zero-shot model outperforms their fine-tuned model by a considerable margin (12.5 vs. 17.44).
>
> Based on the reviewer's suggestion, we conducted an experiment where we fine-tuned our model on the MS-COCO training set and observed an improved FID result of 11.7.
>
> To further demonstrate the superiority of our approach on zero-shot tasks, we reported results on the additional public datasets CUB and LN-COCO (Tab.1). We also demonstrated the advantages of our approach by training all baselines from scratch (including reimplemented DALLE2 and LAFITE) on a customized image-only dataset - 400M stickers collected from the internet (Tab.2). In addition to all these extensive experiments, we analyzed the robustness of our method by applying it to two different diffusion backbones and provided comparisons to each one of them (Tab.2).
>
> **== see also the answer in the next comment ==**
>
> *[1] Nichol, Alex, et al. "Glide: Towards photorealistic image generation and editing with text-guided diffusion models." arXiv preprint arXiv:2112.10741 (2021).* \
> *[2] Saharia, Chitwan, et al. "Photorealistic Text-to-Image Diffusion Models with Deep Language Understanding." arXiv preprint arXiv:2205.11487 (2022).* \
> *[3] Ramesh, Aditya, et al. "Hierarchical text-conditional image generation with clip latents." arXiv preprint arXiv:2204.06125 (2022).* \
> *[4] Gafni, Oran, et al. "Make-a-scene: Scene-based text-to-image generation with human priors." arXiv preprint arXiv:2203.13131 (2022).* \
> *[5] Rombach, Robin, et al. "High-resolution image synthesis with latent diffusion models." Proceedings of the IEEE/CVF Conference on Computer Vision and Pattern Recognition. 2022.*
>
> **[W2-part1] The choice of k**
>
> Thank you for your important comment. For all datasets, we chose k=10 for the number of nearest neighbors. We examined different values of 'k' in the ablation section. As can be seen in the ablation study, the model is not very sensitive to the choice of k, and while k=10 produced the highest FID, any value between 5-12 produces satisfactory results. If you have any other concerns regarding the drawbacks of the retrieval method, please let us know.
>
> Following the reviewer’s question, we have revised the method section to include the value of k (see the end of the retrieval model component).
>
> **[W2-part2] Scaling to larger datasets**
>
> The CLIP image embeddings are stored and extracted using FAISS[1], as described in section 6.5 of the supplement. FAISS is an efficient and powerful tool that is capable of searching billions of data points, hence various works ([2,3,4]) have used it to enable fast kNN search on large datasets. Our study has demonstrated the feasibility of training our model with a large dataset of 400M samples using an inverted file index provided by FAISS. For an efficient search, FAISS assigns Voronoi cells in the d-dimensional space, where each vector falls within a Voronoi cell. Each cell is defined by a centroid, and finding the cell a vector falls in consists in finding the nearest neighbor of the vector in the set of centroids. In search time, only embeddings in the query cell and a few neighboring cells are compared against the query vector. Accordingly, it takes 0.0067+=0.002 seconds to perform a kNN search using our 400M large scale index (Mean and standard deviation calculated on 5000 queries).
>
> Following the reviewer's questions, we have revised the retrieval method section of the supplementary to clarify our use of FAISS. We also added pseudocode that describes the process of constructing the retrieval method. If more clarifications are needed, we will be glad to provide them.
>
> *[1] Johnson, J., Douze, M., & Jégou, H. (2019). Billion-scale similarity search with gpus. IEEE Transactions on Big Data, 7(3), 535-547.*  \
> *[2] Lewis, Patrick, et al. "Retrieval-augmented generation for knowledge-intensive nlp tasks." Advances in Neural Information Processing Systems 33 (2020): 9459-9474* \
> *[3] Karpukhin, Vladimir, et al. "Dense passage retrieval for open-domain question answering." arXiv preprint arXiv:2004.04906 (2020)* \
> *[4] Yalniz, I. Zeki, et al. "Billion-scale semi-supervised learning for image classification." arXiv preprint arXiv:1905.00546 (2019)*

---

> > ### Author Response · Authors · 2022-12-02
> > **A kind reminder to reviewer yR24**
> >
> > Dear reviewer yR24,
> >
> > We would like to thank you again for your time in reviewing our work. As the deadline for discussion is approaching, we really hope to have a further discussion with you. We hope we have addressed your concerns.
> >
> > We believe that we have made significant contributions to several practical problems: (1)  training small and efficient text-to-image models without an explicit paired text-image dataset, (2)  performing local and semantic image manipulations without masks, and without optimization; we present a novel approach that outperforms the recent baselines, and is much faster  (only 8 seconds for each manipulation compared to two hours for Textual Inversion and 9 minutes for Text2LIVE), (3)  zero-shot out of distribution generation
> >
> > We would  be happy to hear your feedback and provide more clarification if necessary.
> >
> > Thank you,
> >
> > The authors

---

> > ### Author Response · Authors · 2022-12-06
> > **Quantitative results for image manipulation**
> >
> > In order to strengthen our benchmarks and to further demonstrate our manipulation capabilities, we conducted an additional human evaluation experiment. In this experiment, the participants were presented with a reference image, a target manipulation prompt, and two alternative results: our result and another baseline result. Participants were asked for their preference, based on (1) semantic similarity to the input image and; (2) alignment to the manipulation text. We performed the survey using a set of 90 pairs consisting of (image, manipulation text). These pairs were selected to maximally cover the set of possible objects and manipulations.
> >
> > The results of the human evaluation showed that the participants preferred our method with regard to both aspects. Specifically, our model outperformed text2LIVE and Textual Inversion in terms of text alignment (our model was preferred 81.6% and 69.3% of the time, respectively) and semantic similarity to the input image (our model was preferred 55.1% and 80.7% of the time, respectively). Unlike the baselines, our model does not require optimization for each input, which makes it much faster during inference.
> >
> > We will include the set of (image, text) pairs and the quantitative results in the final version of the paper.

---

### Author Response · Authors · 2022-11-10
**General response (part 1)**

We thank all the reviewers for their detailed reviews and constructive comments. We are excited to see that they found our work **convincing** (yR24, 5XrX, myk8, o44L), with **strong performance** (yR24, 5XrX, myk8, o44L), **easy to follow** (yR24, 5XrX, myk8) and presenting, among others, a **novel image manipulation application** (yR24, 5XrX).

As we hope we have addressed all of the reviewers' concerns, we will be happy to provide additional clarifications upon request.

Please note that all references to the text and figures refer to the revised version of the paper.

---

### Author Response · Authors · 2022-11-14
**General response (part 2)**


We would like to thank the reviewers again for taking the time to review our paper.

As we are approaching the end of the discussion period, we would be happy to address any remaining concerns per the reviewers' request.

***
The revision parts are highlighted with red text and summarized as follows:

* Added Fig.8 to further demonstrate the importance of our kNN approach for bridging the gap between the images and text distributions. Here, we provide a tSNE visualization of the CLIP image and text distributions when using a different number of nearest neighbors and the cosine similarity score between the distributions.
* Added Fig.11 containing scalability analysis of our model -  here we show that adding kNN to the model consistently improves performance for all model sizes. Furthermore, a performance improvement can be achieved using much smaller models with kNN.
* Revised Fig.3 to reflect reviewers' comments
* Clarified the use of a pre-trained multi-modal text-image encoder when training the model on an image-only dataset.
*  Supplementary: Added a pseudocode for (1) training with kNN conditioning, (2) sampling with kNN conditioning, and (3) retrieval database construction (algorithm 1).
* Supplementary: Added a table listing all the training details (Tab.3)
* Supplementary: Added a background section containing preliminary knowledge on discrete and continuous models (Sec. 6.1).
* Method: Added the kNN condition to the equations and added the number of nearest neighbors.
* Related work: Added semi-parametric models
* Minor and Misc: Revised the paper according to the comments.

*UPDATE* \
To further demonstrate our manipulation capabilities, we have conducted a human evaluation comparison in which we compared our model to several SOTA baselines. The results of the human evaluation showed that the participants preferred our method with regard to similarity to the original object and text alignment (see the second answer to reviewer yR24).


***

***== Contributions ==***

We would like to further clarify the contribution of our paper. Our novel approach tackles the following important and practical problems: (1) training small models without an explicit text-image dataset (but with a pre-trained multi-modal encoder) (2) performing local and semantic manipulations without using masks and without requiring optimization process for each input, (3)  generating out-of-distribution images without fine-tuning and without optimization.

*All these are addressed by our generative model*. In particular:

**(1)** The current SOTA text-to-image models are explicitly trained on large-scale text-image paired datasets. The requirement of paired dataset prevents using the models on many domains where only images are available (e.g. stickers dataset). While both LAFITE and FuseDream tried to solve this problem, LAFITE needed to train an additional model on a text-image dataset in order to achieve better results, and FuseDream required an optimization process for each input.

Our work tackles these issues and proposes a novel approach that **can be applied on top of different backbones, and for each one of them improve their results** (Tab.2). Using extensive experiments (Tab.1-2, Fig.2, Fig.5, Fig.15, Fig.17) we show that leveraging kNN enables training **significantly smaller models** without text-image dataset (using a pre-trained multi-modal encoder), while producing high-quality results that are comparable to much larger models, trained with an explicit text-image dataset.

**(2)** Current manipulation techniques are either (a) limited to global editing; (b) rely on masks; (c) don’t preserve identity; or, (d) require optimization for each input.

We propose a novel approach to perform semantic and local manipulations, **without using masks, and without any optimization**. Our generation process time is **only 8 seconds**. This is a significant improvement compared to the recent Textual inversion which takes 2 hours and Text2Live which takes 9 minutes. In addition, compared to the baselines, our model is capable of performing challenging manipulations, while preserving the identity of the object (Fig. 4, Fig.20-22). This observation is also demonstrated using human evaluation experiment.

**(3)** We demonstrate out-of-distribution generation capabilities **without fine-tuning** (Fig.7, Fig.9). Using a model trained with a retrieval method on a specific index, we demonstrate how one can generate images of different distributions and styles that the model was not trained on, by changing only the index, without fine-tuning.

---

> ### Author Response · Authors · 2022-11-17
> **We would be happy to address any follow-up questions**
>
> Dear reviewers,
>
> Thank you again for your detailed feedback and useful ideas that greatly improved our manuscript.
>
> We would respectfully like to follow up to see if our response addresses your concerns. We believe that our reply fully addresses all raised concerns and would appreciate the opportunity to discuss our work further if the response has not already addressed all concerns.
>
> Thank you,
>
> The authors

---

### Decision · Program_Chairs · 2023-01-20

**Decision:**

Accept: poster

**Justification For Why Not Higher Score:**

Overall the reviewers agree that this paper presents an interesting approach for bridging between parametric and non-parameteric approaches for text-to-image diffusion models. The experimental results support the claims and present surprising strong results at times. However, given the concerns regarding the clarity of writing and missing experimental details, the reviewers recommended the poster acceptance.

Final note: The reviewers in the AC-reviewer meeting brought up a potentially concurrent work [1] that also builds on top of retrieval ideas in diffusion models. However, given the concurrency of this work, the AC and reviewers decided not to penalize the current submission for novelty threats from the concurrent works. **However, they all agree that the related work [1] should be discussed in details in the related work section of this submission.** The authors are strongly encouraged to discuss this work and contrast their method against it in the camera-ready version.

[1] Semi-Parametric Neural Image Synthesis Blattman et al., https://arxiv.org/abs/2204.11824v3

**Justification For Why Not Lower Score:**

Given the potential impact of KNN diffusion models, the reviewers and AC believe that the paper should be given a chance to be presented at ICLR.

**Metareview: Summary, Strengths And Weaknesses:**

This paper proposes an interesting approach for using a set of K nearest embeddings for text-to-image generation using diffusion models. The basic rationale is that CLIP image embedding may not be well aligned with CLIP text embeddings. So, for text-to-image generation, rather than specifying a conditioning input using text embedding of the input prompt, one can use K nearest image embeddings as input to the diffusion model. This paper bridges the gap between fully parametric and non-parametric models and shows that KNN diffusion models can achieve interesting results using small models.

Pros:
- An interesting approach has been proposed for introducing non-parametric components into text-to-image diffusion models.
- The paper archives empirically strong results using small models

Cons:
- Most reviewers have found many missing details in the original submission (ranging from evaluation protocol to training and architecture details). Some of these details are added during the rebuttal period.
- Writing and presentation could be improved.

**Note From Pc:**

if the above contains the word "oral" or "spotlight" please see: "oral" presentation means -> notable-top-5% and "spotlight" means -> notable-top-25%. As stated in our emails, we are disassociating presentation type from AC recommendations

**Summary Of Ac-Reviewer Meeting:**

3 out of 4 reviewers attended the AC-reviewer meeting including o44L, 5XrX, and yR24.

5XrX made the following points:
- results are convincing (+)
- the idea is not very original but it is interesting and surprising to see that having neighborhood helps so much in generation (+)
- the major concern is that the paper claims not to use text-image pair but this was corrected in the rebuttal (+)
- Many missing details in the original submission (-)

o44L made the following points:
- Writing is rushed, many details are missing but the rebuttal process improved the paper significantly (+)
- Novelty is not strong but enough, the results are solid enough, KNN is a strong baseline in the industry and in practice (+)

yR24 mentioned:
- It is very interesting to consider retrieval in generation (+)
- The paper could discuss the retrieval components and their impact on generation more (-)
- Missing details (-)
- The performance seems strong in the controlled setting (+)


All the reviewers present in the meeting agreed that the paper can be a good poster paper at ICLR. However, they do not recommend the spotlight option due to some of the major missing details that were added in the rebuttal period.